# ArcMemo: Abstract Reasoning Composition with Lifelong LLM Memory

## Abstract

While inference-time scaling enables LLMs to carry out increasingly long and capable reasoning traces, the patterns and insights uncovered during these traces are immediately discarded once the context window is reset for a new query. External memory is a natural way to persist these discoveries, and recent work has shown clear benefits for reasoning-intensive tasks. We see an opportunity to make such memories more broadly reusable and scalable by moving beyond instance-based memory entries (e.g. exact query/response pairs, or summaries tightly coupled with the original problem context) toward **concept-level memory**: reusable, modular abstractions distilled from solution traces and stored in natural language. For future queries, relevant concepts are selectively retrieved and integrated into the prompt, enabling test-time continual learning without weight updates. Our design introduces new strategies for abstracting takeaways from rollouts and retrieving entries for new queries, promoting reuse and allowing memory to expand with additional experiences. We evaluate on ARC-AGI, a benchmark that stresses compositional generalization and abstract reasoning, making it a natural fit for concept memory. Our method yields a **7.5%** relative gain over a strong no-memory baseline, with performance continuing to scale with inference compute. We find abstract concepts to be the most consistent memory design, outscoring the baseline at all tested inference compute scales. Moreover, dynamically updating memory during test-time outperforms fixed settings, supporting the hypothesis that accumulating and abstracting patterns enables further solutions in a form of self-improvement.

## 1 Introduction

Large language models (LLMs) have made substantial strides in reasoning-intensive tasks with long-form reasoning. However, a notable opportunity lies in the fact that LLM systems are fixed after deployment: new patterns and strategies uncovered during deep reasoning are not yet carried forward once the context is cleared. This contrasts with human approaches to solving complex, compositional reasoning problems, which involve building on prior insights, abstracting patterns, and composing them in new contexts. Augmenting LLMs with memory offers a natural solution to retaining and building on their discoveries.

While external augmentation was initially designed for factuality in knowledge-intensive tasks (e.g. Lewis et al. (2021) with RAG, Zhang et al. (2019) with Knowledge Graphs, *inter alia*), recent work has begun developing memory approaches for reasoning-intensive tasks, storing questions, findings, and mistakes in memory instead of simple facts (Yang et al., 2024). These memories tend to be specific to the problem/experience it was originally derived from (we term these "instance-level" concepts, see Figure 1). Although effective for closely related problems, they have diminished utility for problems superficially different from prior experiences. Other approaches, such as Suzgun et al. (2025), begin addressing the "instance-level" issue by maintaining an evolving summary of previous experience to great effect, but the lack of structure and modularity makes scaling with more experiences challenging. Without selective retrieval, the size of memory is limited, and the problem solving model has the additional burden of picking out the relevant ideas from all past experiences.

We introduce an *abstract concept-level memory* framework to support compositional reasoning for all subsequent queries. We emphasize (1) **abstract concepts** that are more general and separated from their original context to be useful across a larger set of future problems, and (2) **modular**

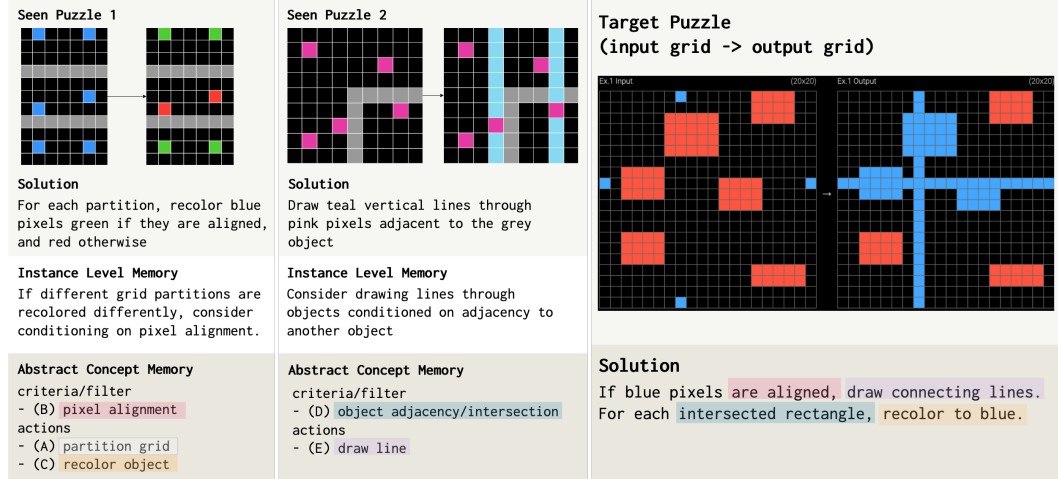

Figure 1: **Instance-Level vs. Abstract Concepts Example.** Each ARC-AGI Chollet (2019) puzzle requires inferring the transformation rule for a set of input/output pixel grids. Here, Puzzle 1 instantiates $(A \wedge B) \Rightarrow C$ and Puzzle 2 instantiates $D \Rightarrow E$. The target puzzle is solved by recombining these ideas ($B \Rightarrow E, D \Rightarrow C$). Instance-level memory tends to store fully composed rules, coupling $A$ with $B, C$, and so on. Transferring to the target then demands both ignoring $A$ and disentangling/reordering $B, C$ with $D, E$. Abstract memory instead stores $A, B, C, D, E$ as separate, modular concepts, making them easier to recognize and reassemble in new contexts.

**concepts** that directly promote recombination with other ideas that can be easily built on with new experiences or curated for a target problem. We distill solution traces (both from the system itself or from external sources) into reusable, modular abstractions stored in natural language. At inference time, relevant concepts are selected and integrated into context, enabling test-time continual learning without expensive weight updates. Our design encapsulates two primary operations: (i) writing to memory: formatting concepts for abstraction and generality; and (ii) reading from memory: selecting a subset of concepts for the current problem. We title our method **ArcMemo— A**bstract **R**easoning **C**omposition **Memo**ry.

As illustrated in Figure 1, instance-level memories often capture an entire solution pattern tied closely to its original problem (e.g., the joint use of ideas $A, B, C$ in Seen Puzzle 1). Such overspecified entries are less likely to recur in future problems, and even when partially relevant to the Target Puzzle, the agent must still disentangle useful pieces like $B$ from the original bundle. In contrast, abstract concepts are stored individually with fewer contextual assumptions, making them easier to recognize, adapt, and recombine across superficially different puzzles.

We evaluate on ARC-AGI, where simple pixel-grid operations compose into a vast task space, and solving tasks requires compositional reasoning rather than memorizing individual patterns, making it a natural testbed for ArcMemo. ARC-AGI simultaneously remains challenging for SOTA models and solvable by human children. Along with clean, deterministic scoring, this reasoning benchmark simulates frontier domains not present in vast training corpora and thus serves as a perfect measure for a continual learning system. On ARC-AGI-1, our method improves the official score from 55.17 to 59.33 (+7.5% relative gain over a strong no-memory baseline) and is the only evaluated memory design we find to outperform the baseline at all inference scales. Our experiments confirm that continually updating memory at evaluation time yields a score improvement that emerges with inference scale via retries: memory updates triggered from a previous pass over the test set may enable new solves in a subsequent pass. Finally, we observe that selecting a subset of memories to include for a particular problem improves performance and reduces token cost, indicating the selection operation is essential beyond allowing memory to grow continually.

## 2 RELATED WORK

Our approach draws on several threads of research in augmenting LLMs with memory but contrasts in (1) target applications (reasoning-intensive vs. knowledge-intensive tasks), (2) the underlying

memory modality (text vs. continuous vectors/embeddings), and (3) emphasis on abstraction and modularity for reasoning. We discuss related lines (1) and (2) as well as other related work in Appendix E and emphasize reasoning and abstraction here.

**Memory for Test-Time Reasoning and Concept Abstraction.** More recent work has shifted toward reasoning-centric uses of memory, particularly test-time learning. Think-in-Memory records intermediate reasoning steps as structured triples, storing them in a locality-sensitive hash table to enable reuse in multi-step reasoning tasks (Liu et al., 2023). Buffer of Thoughts stores problem specific reasoning templates, retrieved by embedding similarity and instantiated for new inputs (Yang et al., 2024). It categorizes and summarizes solution attempts into templated insights, which are then added to memory. In contrast, Dynamic Cheatsheet maintains a unified buffer that is adapted continuously. With each problem query, the LLM updates this memory blob by rewriting the entire buffer (Suzgun et al., 2025). Rather than retrieving specific entries, the entire cheatsheet is appended to the prompt, functioning as a persistent cache of problem-solving strategies. We emphasize that our use of memory for long-form reasoning is cross-episodic rather than a working memory for state tracking within a single problem (e.g. Jayalath et al. (2025)'s PRISM).

**ARC-AGI Approaches.** Our work is primarily evaluated against ARC-AGI as a benchmark that simulates complex reasoning on frontier tasks without requiring frontier-level knowledge. Various techniques and findings have been developed against this benchmark. Li et al. (2024) demonstrates the efficacy of synthetic data and the complementary approaches of (i) learning to infer the underlying transformation via program synthesis and (ii) using test-time weight adaptation to directly learn the transformation function in the neural net. Akyürek et al. (2025) further investigates the test-time adaptation method, producing various insights, including that low rank adapters are poorly suited to retaining practical experience across puzzles. Our program synthesis approach can be viewed as a memory-augmented version of Wang et al. (2024)'s hypothesis search or Qiu et al. (2024)'s hypothesis refinement with execution feedback-based retry. Other recent approaches to ARC-AGI include brain-inspired architectural innovations such as Wang et al. (2025)'s hierarchical reasoning model, achieving impressive performance through a recurrent architecture that iteratively refines its output until a halt signal is predicted, together with data augmentation techniques. Finally, the relative ease of generating new ARC-AGI tasks compared to solving them is exploited in Pourcel et al. (2025)'s self-improving system that relabels incorrect programs as the correct program for a different puzzle and trains a model on its generations this way.

## 3 METHODS

### 3.1 PROBLEM STATEMENT

---

**Algorithm 1:** Inference with Continually Updating External Memory

---

**Input:** Dataset $\mathcal{D} = \{x_i, y_i\}_{i=1}^{n}$ (labels $y_i$ optional), External memory $M$, Operations MEMREAD, MEMWRITE, GETFEEDBACK, Update interval $k$,
**Output:** Predictions $\{\hat{y}_i\}$
**for** $i \leftarrow 1$ **to** $n$ **do**
    $(x_i, y_i) \leftarrow$ GETITEM$(\mathcal{D}, i)$ ;    // Label $y_i$ may be absent at inference
    $s_i \leftarrow$ MEMREAD$(M, x_i)$ ;    // Retrieve relevant memory entries
    $\hat{y}_i \leftarrow$ LLM_GENERATE$(x_i, s_i)$ ;    // Predict using LLM + selected memory
    **if** $i \bmod k = 0$ **then**
        $f_i \leftarrow$ GETFEEDBACK$(\hat{y}_i, y_i)$ ;    // test verification, reflection, etc.
        MEMWRITE$(M, x_i, \hat{y}_i, f_i)$ ;    // Incorporate feedback into memory

---

We formulate a memory system as a collection of memory entries associated with read and write operations. Problem solving with memory augmented LLMs thus requires designing three components as seen in algorithm 1:

1. **Memory Format** (3.2): What is stored in individual entries?
2. **Memory Write** (3.3): How is memory updated from a reasoning trace?
3. **Memory Read** (3.4): How is memory used for tackling new problems?

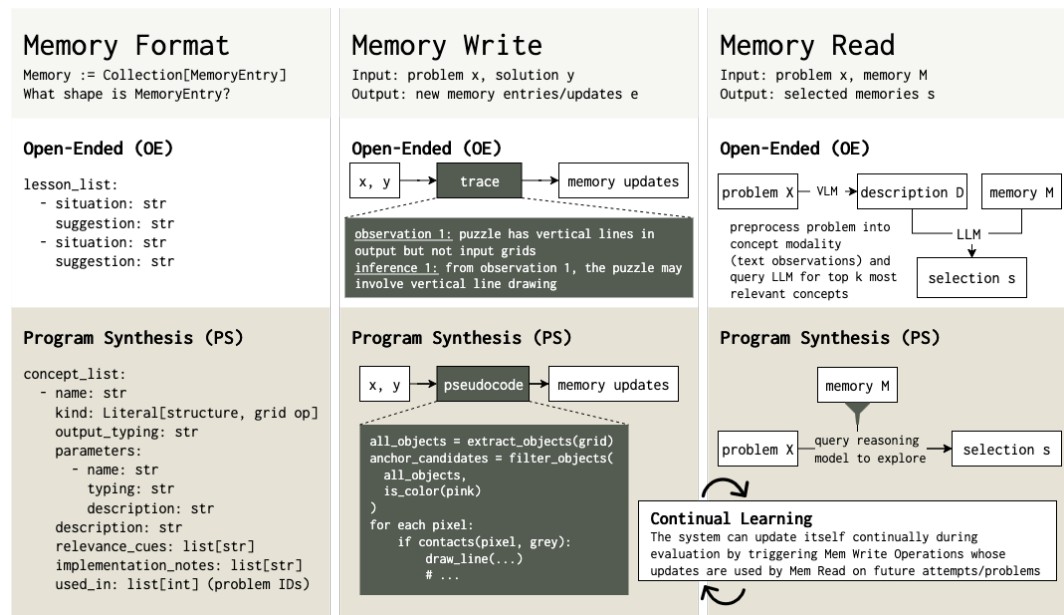

Figure 2: **Method Diagram.** Implementing a memory system requires defining (1) what is stored, (2) how memory is updated, and (3) how memory is used for new queries. The key novelty in this work is emphasizing abstraction and modularity, and the corresponding design changes. In particular, we highlight that parameterization (with higher-order functions allowed and encouraged) promotes abstraction, and typed interface definitions support modularity by showing which concepts can be combined. Since these memory entries are more abstract, they also require more inference to map against new, concrete situations–whether by aligning input against the memory format in a preprocessing query, or leveraging reasoning models to explore in a directed manner.

These are generic design considerations for any memory. Our method's novelty lies in the choices made to target (1) abstracting a memory to be less specific to the problem it was derived from (2) enabling memory selection, since greater abstraction makes standard retrieval approaches (embedding similarity thresholding) less effective. We present two such implementations: (1) an open-ended (OE) format that imposes minimal constraints on entry format and uses a simple problem preprocessing step for memory selection, and (2) a program-synthesis (PS) inspired format that categorizes and parameterizes concepts and uses reasoning model exploration to select relevant items.

**Problem Assumptions.** The main requirement for our memory system is that some feedback is available at test-time (e.g. via test cases or self-reflection). This condition is necessary as retaining patterns from an incorrect or otherwise flawed trace would serve to carry mistakes forward. In other words, some signal is needed to discriminate correct from incorrect traces to ensure only productive ideas and patterns are added to memory. Incorrect traces still likely contain vital signal, but error identification/credit assignment remains an open challenge that we leave to future efforts. Various real-world tasks may satisfy this requirement, as tasks are often specified with either examples showing desired behavior or evaluation criteria defining what outcomes are positive and negative. To give some examples, code completion tools have criteria like compilation success and test pass rate, and medical diagnosis tools have evaluation criteria defined as patient outcomes (e.g., survival rates). In the case of our evaluation benchmark ARC-AGI, puzzles are explicitly presented as a set of input-output examples, which, for our purposes, can serve as automatically verifiable test cases.

## 3.2 MEMORY FORMAT AND ORGANIZATION

Specifying an individual memory entry's format involves selecting fields useful for retrieval and downstream problem-solving. The organization of memory entries on a collective level is another design surface we leave for future exploration; we use a flat collection of entries for simplicity.

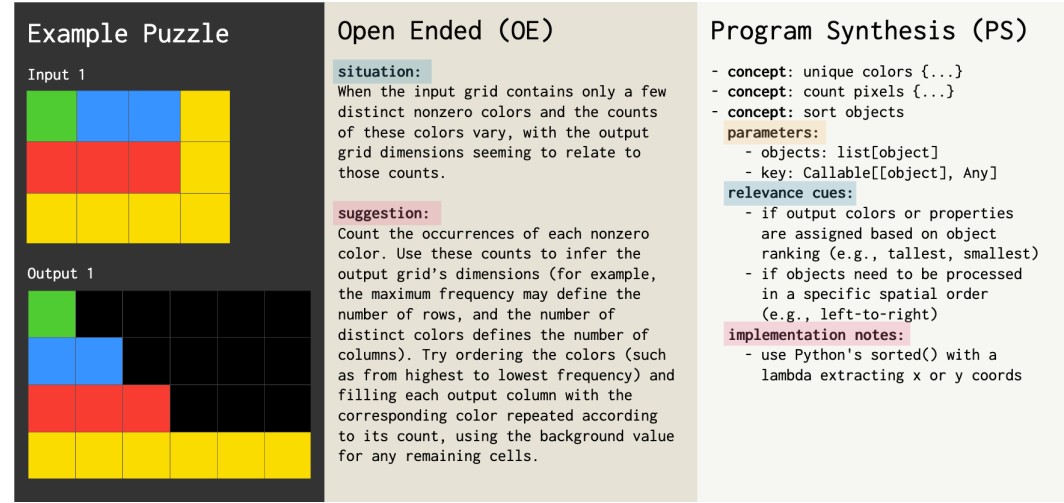

Figure 3: **Open-Ended (OE) vs. Program Synthesis (PS) Concept Examples.** An example of concepts from puzzle `9af7a82c` abstracted into each concept format. OE defers to the model, while PS imposes structure to encourage abstraction/modularity. Higher order behavior is demonstrated with the "sort objects" concept taking a `Callable` parameter that specifies specific variations.

**Open-Ended (OE) Formulation.** We first consider an open-ended approach that imposes minimal structure and defers formatting individual memories to the model. Following the basic lesson format of "under situation X, do action Y", the only formatting constraint we impose is having distinct `situation` (X) and `suggestion` (Y) fields. This separation is designed for abstraction (disentangles the core idea from the original context) and modular reuse (`situation` explicitly describes the conditions where this idea can be reused).

**Program Synthesis (PS) Formulation.** In early iterations of automatic concept summarization from solution traces, we observed outputs to occasionally be overly specific. Consider Figure 3's OE example that bundles ideas of counting, drawing columns, and sorting into a single entry. This overspecification inherits the same limitations as instance-level concepts: friction when matching a composite idea to new scenarios and redundancy from composite ideas sharing components.

We designed a more structured alternative to the OE format to address these issues and more strongly enforce the ideals of abstraction and modularity. In particular, we took inspiration from software engineering and functional programming, which respectively have existing solutions in modular design for code reuse and higher-level functions for composition. From this analogy, our Program Synthesis (PS) format frames concepts as types, structures, and routines. The primary feature is that each of these concepts is parameterized. Parameterization allows similar concepts to be represented compactly with variations abstracted into parameters. Moreover, type annotations for inputs (parameters) and outputs (return types) encourage modularity and composition: the typed interfaces suggest what ideas fit together and how. Finally, by allowing parameters to be routines themselves (introducing higher-order functions), we encourage further generalization by enabling abstraction over specialized logic and routines rather than just values. Higher-level patterns can be recognized across instances and stored with routine arguments defining the different lower-level logic. A subtle benefit of this more structured formulation is that memory representation can be easily compressed by omitting certain fields (see subsection A.1 for a complete list of PS concept fields).

### 3.3 MEMORY WRITE: CONCEPT ABSTRACTION

Memory's primary purpose is to persist discoveries and reflections from prior experiences. Converting experiences to memory updates is then the most crucial operation for achieving persistence.

**OE Abstraction.** Abstracting suggestions from a solution trace is straightforward–simply query a model to reflect on the solution trace and summarize specific general ideas that may be reused

for future puzzles. To synthesize specific conditions or situational cues to pair with suggestions, ideally, we can directly refer to the original problem-solving process as a series of deductions that derive new facts or explore ideas from existing facts, and link each suggestion (idea) with the prior observations that inferred this suggestion. However, intermediate inferences may be implicit or difficult to extract from a trace, or, in the case of specific commercial models, completely hidden. To address this challenge, we generate a post-hoc derivation: an interleaved sequence of observations and thoughts/reasoning steps constructed from the input-output of the final solution. These reconstructed traces are then used to extract situation-suggestion pairs, forming structured memory entries for guiding future analyses. This minimal formulation does not consider the existing contents of memory while adding new entries for simplicity and scaling reasons. It defers handling redundancy and consolidating ideas to the concept selection and problem-solving phases, respectively.

**PS Abstraction.** We find that directly converting solutions into routines often leads to minor implementation details being recorded as memories, as opposed to our intended abstract concepts. We mitigate this by preprocessing solutions into pseudocode to prioritize higher-level operations over low-level implementation details. The main abstraction phase proceeds from the generated pseudocode, recording new concepts and revising existing concepts along with their various fields.

In the spirit of designing for abstraction, promoting reusability, and minimizing total concept memory length, this operation *is* aware of existing concepts with a compressed form of memory included in context. We encourage the model to reuse and revise existing concepts by updating concept descriptions, parameter lists, relevance cues, and implementation notes. Importantly, we encourage higher-order functions with routines as arguments in instructions and few-shot demonstrations.

All concept abstraction/memory writing operations are scaffolded with few-shot demonstrations, example-rich templates, and comprehensive instructions.

## 3.4 MEMORY READ: CONCEPT SELECTION

Although state-of-the-art models support longer context windows, their maximum length still limits how much information or how many memory entries can be included in a single query. Even without this hard limit, including all available memories in context is still undesirable: it can distract the model with irrelevant details and flood it with too many unlikely hypotheses to consider. A more effective strategy is to introduce a selection mechanism only to include the most relevant subset of memory entries at problem-solving time. This allows the memory store to grow continually without overwhelming the model's context window.

**OE Selection.** OE-format memory entries were explicitly designed to support memory-based selection. The `situation` clause acts as a semantic hook to identify relevance. The key idea for OE Selection is a preprocessing step that leverages model reasoning capability to parse the abstract domain input into problem descriptions at different levels of abstraction to facilitate matching against our abstract concepts. Since ARC-AGI deals with spatial reasoning, we leverage a vision language model for this preprocessing step. We caption each puzzle using a structured prompt that separates concrete observations from speculative transformations. This converts spatially rich input into a natural language format suitable for matching against stored situations. We then query a model for the top-$k$ most relevant entries using the generated description. We explored scoring, thresholding, and cumulative similarity approaches (as in top-$p$ sampling). Observing comparable results, we selected top-$k$ cutoff for simplicity. We consider this a light-weight approach to converting abstract domain inputs into more understandable and memory-aligned representations.

**PS Selection.** Standard embedding approaches use a single forward pass per embedding, representing a System 1-style fast thinking/intuition-driven understanding from the model. Intuitions may not be sufficient for difficult/frontier tasks and domains. Moreover, the connection between a higher-level concept and a specific problem instance may not be immediately clear because of the abstraction.

To address these challenges, we propose **reasoning-based selection**–using System 2-style thinking to deliberately think and explore. In contrast to the more straightforward playbooks curated in the OE memory format, the explicitly abstract PS concepts are a set of puzzle pieces with notes on identifying if a piece is relevant (relevance cues) and on how various pieces fit together (type annotations). To select from this memory, we propose leveraging model-driven exploration instantiated by

recent models' long-form reasoning with backtracking. PS Selection instructs a reasoning model to systematically explore the problem: first identify initial concepts using relevance cue annotations, then attempt to "fill in the details" by determining values or routines to populate these initial concepts' parameters using type annotations to identify which other concepts should be investigated.

## 3.5 CONTINUAL CONCEPT LEARNING

A long-held goal in machine learning is to develop lifelong learning systems that continually self-improve without manual intervention. A memory system that can leverage the learning signal present at test-time via continual updates is one approach to achieve this ambition. Our memory write operations are lightweight queries that can ingest solution traces derived from both the system and external sources, making continual updates practical at scale.

However, continual updates also introduce dependencies on evaluation order. If solving problem $i$ induces a memory update that enables problem $j$ to be solved, then performance differs between $(..., x_i, ..., x_j, ...)$ and $(..., x_j ..., x_i, ...)$. Inference batching further complicates this: even if $x_i$ precedes $x_j$, they may appear in the same batch, so the model attempts $x_j$ before $x_i$ has updated memory. This introduces an accuracy–throughput trade-off. In all settings, we initialize our external memory with seed data (problems and solutions). We explicitly evaluate continual updates in subsection 5.3's experiments, confirming their efficacy. In other experiments, we used fixed memory to prioritize throughput and avoid the potentially confounding effect of order dependencies.

## 4 EXPERIMENTS

**Benchmark Selection.**    We evaluate our proposed framework on ARC-AGI-1 (Chollet, 2019), a benchmark explicitly designed to evaluate intelligence as "efficient acquisition of new skills" instead of "fixed possession/memorization of skills." Each ARC puzzle encodes a transformation rule that maps input to output pixel grids. The objective of each puzzle is to infer its rule given several examples of input-output grid pairs, and produce the corresponding output grid for several input test cases. We find that the abstract domain of pixel grid transforms provides a meaningfully challenging testbed to simulate frontier domains without requiring expert knowledge to evaluate trajectories–a confluence of desirable properties for evaluating a continual concept learning system.

ARC-AGI-1 contains a public validation split containing 400 puzzles with a difficulty distribution matching that of the private evaluation. Following Akyürek et al. (2025), we evaluate a randomly selected 100-puzzle subset of the public val split. This makes repeated runs for more stable estimates feasible given cost and the sampling variance we observed. Li et al. (2024) manually authored Python solutions for 160 puzzles from the public train split to act as "seeds" to recombine into their synthetic dataset. We reuse these solutions to seed our memory rather than training data.

**Models.**    To build on frontier models, we experiment primarily with OpenAI's o4-mini. At the time of writing, o4-mini is second only to Grok 4, but o4-mini's lower price puts it on the Pareto frontier of cost and performance ARC-Prize (2025). For auxiliary tasks such as concept abstraction and non-reasoning selection, we use OpenAI's GPT-4.1 to conserve token usage. Early experiments also evaluated the open-weight DeepSeek R1, which has the benefit of visible thinking traces, but its 8000 output token limit consistently yielded unfinished solutions in initial testing.

**Evaluation.**    While the official evaluation harness queries models to directly predict output grids for test cases, we instead use a program synthesis approach that queries for a transformation function to convert input to output grids. The code artifact provides more signal for reflection and also allows us to test proposed logic against reference pairs for feedback. We evaluate performance under 0, 1, and 2 retries with this execution feedback (we observed diminishing returns with further retry, and choose this threshold to conserve cost). We follow official ARC-AGI scoring (two attempts per puzzle), and account for sampling variance by averaging over extra runs (see details in Appendix B). The primary memory baseline we compare against is a re-implementation of Suzgun et al. (2025)'s DC-Cu (labeled "cheatsheet") that uses a frozen memory to match other settings.

# 5 RESULTS

## 5.1 MAIN RESULTS

| Setting | Oracle@1 | **Oracle@2 (Official)** |
|---|---|---|
| qwen3-235b-a22b-instruct | 11.00 (1.00) | 16.67 (0.57) |
| deepseek r1 | 19.50 (7.05) | 26.33 (4.02) |
| claude sonnet 4 (thinking 16k) | 45.50 (1.00) | 54.17 (0.58) |
| gemini 2.5 flash (thinking 16k) | 9.67 (4.86) | 13.83 (4.73) |
| o4-mini (medium) | 46.33 (1.04) | 55.17 (3.18) |
| cheatsheet (o4-mini (medium)) | 47.50 (2.78) | 57.67 (2.52) |
| ArcMemo-OE (ours, o4-mini (medium)) | 48.00 (1.00) | 56.67 (1.53) |
| **ArcMemo-PS (ours, o4-mini (medium))** | **49.33 (0.29)** | **59.33 (0.29)** |
|   + one retry | 58.00 (2.29) | 67.33 (1.61) |
|   + two retries | 61.67 (3.88) | 70.83 (3.06) |

Table 1: **Main Results.** ArcMemo-PS yields the best results and scales with additional compute. Bracketed values represent standard deviation. See additional score details in Table 4.

As seen in Table 1, ArcMemo-PS achieves the best official score with standard compute [1] and still benefits from additional scaling in both parallel samples and sequential retries. Table 4 shows complete details on all settings at all scales, where ArcMemo-PS is the only setting that consistently outperforms the baseline in all inference compute scales. Other memory formulations (Cheatsheet, ArcMemo-OE) situationally improve over the baseline but still underperform it in some regimes.

We observe that ArcMemo-PS's advantage is most prominent with lower compute regimes. This is consistent with the stated goal "memory for reasoning": persisting previous experience aims to reduce redundancy from rediscovering ideas found in previous rollouts. The marginal impact of memory is reduced with more inference compute as the model can rediscover ideas through exploration.

Compared to our memory baseline (Cheatsheet), ArcMemo methods are highly competitive, performing favorably in most regimes. The quantitative improvement shown by ArcMemo is supported by qualitative analysis that demonstrates ArcMemo memories are more modular. This qualitative analysis (described more in subsection 5.2) observes that these abstract memories improve concept coverage–more target puzzle ideas are reflected in memory. This difference may be explained by ArcMemo's modular design being more conducive to maintaining a growing library.

## 5.2 SELECTION ABLATION AND QUALITATIVE RELEVANCE ANALYSIS

| | | Oracle@k Scores | | |
|---|---|---|---|---|
| retry | setting | k=1 | **k=2 (Official)** | k=3 |
| 0 | ArcMemo-PS | **49.33 (0.29)** | **59.33 (0.29)** | **63.50** |
| | - selection | 46.83 (5.25) | 55.17 (2.02) | 60.00 |
| 1 | ArcMemo-PS | **58.00 (2.29)** | **67.33 (1.61)** | 72.50 |
| | - selection | 57.67 (3.79) | 66.00 (3.00) | **73.00** |
| 2 | ArcMemo-PS | **61.67 (3.88)** | **70.83 (3.06)** | 75.50 |
| | - selection | 61.33 (3.21) | 70.00 (2.65) | **76.00** |

Table 2: **Selection Ablation.** Ablating the reasoning-based selection mechanism from ArcMemo-PS reduces overall performance, showing selection is useful for downstream performance.

We perform an ablation experiment removing the selection mechanism from ArcMemo-PS to examine its impact and to compare more directly to the selection-free Cheatsheet methodology. The generally

---

[1] Matching the official evaluation scheme with 2 parallel attempts and 0 retries.

better performance of the with-selection setting (as seen in Table 2) suggests that, in addition to allowing memory to grow without overwhelming context in both the abstraction and inference phases, it also helps downstream problem-solving performance. Moreover, from the token usage plot in Figure 4, we see that concept selection's performance improvement also comes at higher efficiency–the selection-ablated setting uses far more tokens. However, there still exist certain scale regimes where the no-selection setting has a better score. We attribute this to variance from imperfect selection and note that this indicates headroom for selection/retrieval.

ArcMemo-PS (-selection) is most comparable to Dynamic Cheatsheet's "cumulative" setting (DC-Cu) as all stored memory is included with each query. While both ultimately solve the same number of puzzles across all runs, we find that the settings differ in 10 puzzles (that is, each system solves 5 puzzles the other does not). We manually analyzed these puzzles and found that only 40% of new solves in the cheatsheet setting were related to memory elements actually in the generated cheatsheet. That is, we were unable to find relevant notes in the cheatsheet to explain 60% of its new solves. In contrast, all new solves from ArcMemo-PS (-selection) we examined can be linked to concept memory contents. While we cannot definitively conclude ArcMemo changes directly induced each of the new solves (given the state of LLM interpretability), this seems to suggest new solves in the ArcMemo settings are potentially more attributable to the memory component compared to sampling variance.

Manual inspection of a subset of reasoning-based selection results used for ArcMemo-PS finds that while some irrelevant concepts still appear within the selection, the key idea of the target is often included in the selection. The presence of a few distractor concepts appears to be manageable, as the reasoning model itself is capable of exploring with backtracking.

While concept retrieval generally improves performance, we observe several failure modes. In some cases, the retrieved concepts are irrelevant or overly abstract, lacking the specificity needed for the puzzle. Other times, retrieval succeeds, but the puzzle remains unsolved due to its intrinsic difficulty or because the puzzle requires an entirely unseen or highly detached operation or structure (such as extending `Draw lines in a spiral form` from the retrieved `Draw line`). Notably, there are instances where relevant concepts (e.g., hole counting) exist in memory but are not selected—suggesting possible perception or alignment gaps. These cases highlight headroom for improving both the selection model and downstream reasoning robustness.

## 5.3 Continual Learning

| retry | setting | Oracle@k Scores | | |
|---|---|---|---|---|
| | | k=1 | **k=2 (Official)** | k=3 |
| 0 | ArcMemo-OE | 48.00 (1.00) | 56.67 (1.53) | 61.00 |
| | + continual memory update | 46.33 (1.53) | 56.00 (0.00) | 61.00 |
| 1 | ArcMemo-OE | 56.67 (2.08) | 65.67 (1.53) | 70.00 |
| | + continual memory update | 57.17 (3.69) | 65.00 (0.00) | 67.00 |
| 2 | ArcMemo-OE | 60.67 (1.53) | 67.67 (2.52) | 71.00 |
| | + continual memory update | **62.33 (3.51)** | **70.00 (1.73)** | **72.00** |

Table 3: **Continual Memory Learning.** Comparing otherwise identical memory systems, we find that with sequential inference compute scale (at high retry depth), continually updating memory with new self-generated solutions leads to improved puzzle-solving performance.

Results from our experiment comparing ArcMemo-OE with a version that updates *during* evaluation (every 10 problems; Table 3) show that our memory system's performance improves with continual updates–a key requirement for meaningful lifelong learning. In particular, we find that the performance improvement emerges at later iterations in sequential inference scaling. We hypothesize this is because, after more iterations and puzzle-solving passes, new solutions are found, new memories are abstracted, and these new memories help solve other puzzles.

## 5.4 ADDITIONAL RESULTS

**Expanded Scale.** Evaluating ARC-AGI methods is costly due to the long reasoning traces and high sampling variance required for reliable estimates. Following prior work, our main experiments therefore use a representative subset of the validation set. For completeness, we additionally evaluate our ArcMemo-PS configuration on the full public validation split. As shown in Table 6, ArcMemo-PS outperforms the baseline at every retry level and contributes improvements on 21 puzzles overall.

We also expand the set of tested models to DeepSeek R1, Sonnet 4, and Kimi K2 Thinking (Table 7). Using the same GPT-4.1-derived memory, R1 improves from 26.33 to 30.67 (+16%), and K2 from 37.33 to 40. With Sonnet 4 as the base model, ArcMemo still improves on 7 puzzles over the baseline. These results indicate meaningful cross-model transfer, and suggest that concept memories distilled from stronger generators (e.g., GPT-4.1) provide useful signal even for models with different architectures. The mixed result on Sonnet 4 further suggests that strong models may benefit less from concepts abstracted from weaker models.

**Other Domains and Intrinsic Self-Improvement.** Although ARC-AGI is a natural setting for continual learning, its tasks are synthetic by design. To test generality, we evaluate on math reasoning using the more portable OE configuration. On the 60 problems from AIME '24 and '25, concept memory improves Gemini-2.5-Flash-Lite-Thinking from 43 to 48 solved problems (+11.6%), demonstrating transfer beyond the ARC domain. ARC intrinsically enables correctness estimation via reference input-output grids as test cases, allowing our method to abstract concepts from "correct" traces, however various task settings provide less signal or feedback. Preliminary AIME results using self-reflection to abstract from all traces rather than only "correct" traces yields 47 solves (+9.3%), suggesting ArcMemo may be effective even in sparse-feedback regimes.

**External Solution Ablation.** Our main experiments initialize the memory with human-written BARC solutions to reduce inference cost. To evaluate a more realistic "pure self-improvement" regime, we instead initialize memory using fully self-generated trajectories. This variant slightly outperforms the original setup (60.67 vs. 59.33). We hypothesize that this on-policy memory abstraction produces concepts that align more closely with the model's inductive biases.

**Pseudocode Preprocessing Ablation.** Our pipeline includes a trajectory-preprocessing step that converts solutions into pseudocode before abstraction. Early tests suggested this step improves generality; we confirm this with a direct ablation. Removing pseudocode preprocessing reduces the ArcMemo-PS score from 59.33 to 55.67—only marginally above the baseline. Since this step is inexpensive (no deep reasoning required) yet materially improves abstraction quality, we retain it in the full system.

We further discuss token efficiency, concept specificity, and embedding retrieval in Appendix C, and limitations and future work discussion in Appendix D.

## 6 CONCLUSION

In this work, we introduce ArcMemo, a framework designed to support lightweight, lifelong learning on compositional reasoning tasks by emphasizing a higher level of abstraction and modularity. We explore two implementations of memory modules against ARC-AGI, a benchmark specifically designed to resist memorization and to evaluate fluid intelligence instead. Our main findings confirm the efficacy of our approach: ArcMemo-PS outscores comparable methods under the official evaluation protocol and continues to benefit from additional inference compute scale. Moreover, we observe that continual updates benefit memory augmentation over multiple attempts and that selecting a subset of memory for each problem is a crucial component to enable a memory to continually grow without overwhelming the LLM context. This paper's work represents early attempts toward the main tenets of higher-level abstraction and modularity. Promising future directions include hierarchical designs and consolidation mechanisms that restructure memory. To encourage further investigation, we also release a concept-annotation dataset and configurable puzzle synthesis pipeline, providing resources for evaluating concept representations and advancing abstraction-based memory methods.

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

# A IMPLEMENTATION DETAILS

## A.1 PS MEMORY FORMAT

Each PS format memory entry contains:

- **Title**: a succinct label for the underlying idea.
- **Description**: elaboration on behavior and role.
- **Kind**: whether this concept encodes a type/structure/routine.
- **Parameters**: a list of fields that parametrize this concept.
- **Output Typing**: specifies what the output for this routine is, to suggest how various routines can plug into other routines.
- **Relevance Cues**: much like the `situation` field in the OE concepts, we consider the context in which this concept is relevant.
- **Implementation Notes**: suggestions on how to implement the concept in actual code.

## A.2 EXPERIMENT PARAMETERS

To build on frontier models, we experiment primarily with OpenAI's o4-mini (max tokens=32000, reasoning_effort=medium). For auxiliary tasks such as concept abstraction and non-reasoning selection, we use OpenAI's GPT-4.1 (temperature=0.3, max tokens=1000) to reduce token usage. We looked into evaluating the open-source DeepSeek R1, which also has a transparent thinking process, but the output 8000 token limit consistently yielded unfinished solutions.

# B EVALUATION DETAILS

We use ARC-AGI's official scoring metric, which we term oracle@$k$ (each test case is scored separately, and if any of $k$ candidates pass, full credit is given for that test case). The ARC-AGI official evaluation has $k = 2$, but to mitigate sampling variance, we sample 3 runs for each setting and report the average single run score, average oracle@2 score, and their standard deviation. We investigate several alternate scoring protocols (requiring a single program to solve all test cases, accumulating test case solve rates, and requiring references also to be solved) and include the results in Table 5.

Here is a precise definition of our scoring procedure for a single problem $P$. Let...

- $X$ be the size $n$ set of attempts on problem $P$
- $T$ be the set of test cases for problem $P$
- $C$ represent a particular $k$-subset of $X$, in notation: $C \in [X]^k$

$$score = \frac{1}{|[X]^k|} \sum_{C \in [X]^k} z(C)$$

where $z$ is the problem score by a $k$-attempt ensemble

$$z(C) = \frac{1}{|T|} \sum_{t_i \in T} \mathbb{1}\{\exists c \in C \ \texttt{Verify}(c, t_i)\}$$

where

- $c$ is a single program attempt in the k-attempt ensemble,
- t is a test case,
- and `Verify` returns true if the program successfully passes the test case.

| | | Individual Run Scores | | | Oracle@k Scores | | |
|---|---|---|---|---|---|---|---|
| | | run0 | run1 | run2 | k=1 | **k=2** (Official Score) | k=3 |
| iteration | setting | | | | | | |
| 0 | baseline | 46.00 | 45.50 | 47.50 | 46.33 (1.04) | 55.17 (3.18) | 59.50 |
| | cheatsheet | 48.00 | 44.50 | 50.00 | 47.50 (2.78) | 57.67 (2.52) | **64.00** |
| | ArcMemo-OE (ours) | 49.00 | 47.00 | 48.00 | 48.00 (1.00) | 56.67 (1.53) | 61.00 |
| | ArcMemo-PS (ours) | 49.50 | 49.00 | 49.50 | **49.33 (0.29)** | **59.33 (0.29)** | 63.50 |
| 1 | baseline | 57.00 | 55.50 | 61.00 | 57.83 (2.84) | 66.67 (3.82) | 71.50 |
| | cheatsheet | 56.00 | 56.50 | 57.50 | 56.67 (0.76) | 65.67 (1.44) | 70.50 |
| | ArcMemo-OE (ours) | 59.00 | 55.00 | 56.00 | 56.67 (2.08) | 65.67 (1.53) | 70.00 |
| | ArcMemo-PS (ours) | 58.50 | 60.00 | 55.50 | **58.00 (2.29)** | **67.33 (1.61)** | **72.50** |
| 2 | baseline | 59.50 | 59.00 | 65.00 | 61.17 (3.33) | 69.00 (2.65) | 73.00 |
| | cheatsheet | 65.00 | 61.00 | 61.00 | **62.33 (2.31)** | **71.33 (1.53)** | **76.00** |
| | ArcMemo-OE (ours) | 62.00 | 59.00 | 61.00 | 60.67 (1.53) | 67.67 (2.52) | 71.00 |
| | ArcMemo-PS (ours) | 60.50 | 66.00 | 58.50 | 61.67 (3.88) | 70.83 (3.06) | 75.50 |

Table 4: **Full o4-mini Results.** The right partition of the table containing aggregate scores shows the inference scaling: going down represents sequential retry, going right represents adding parallel samples. ArcMemo-PS is the only evaluated setting that outperforms the baseline consistently at every tested scale.

# C FURTHER DISCUSSION

**Token Efficiency** We find that our system's token efficiency is similar to the baseline. We observe that memory-augmented runs tend to increase output token usage. While this seems to counter the intuition that retrieving rather than regenerating certain ideas would save tokens, we hypothesize that introducing memory leads the model to explore more hypotheses. In other words, selection inaccuracies (false positives) seem to be manageable by the reasoning models, but at the cost of increased token usage. Full results are plotted in Figure 4.

| iteration | setting | run0 | run1 | run2 | k=1 | k=2 | k=3 |
|---|---|---|---|---|---|---|---|
| 0 | baseline | 46.00 | 45.00 | 47.00 | 46.00 (1.00) | 54.67 (3.21) | 59.00 |
| | cheatsheet | 48.00 | 44.00 | 50.00 | 47.33 (3.06) | 57.67 (2.52) | 64.00 |
| | ArcMemo-OE | 49.00 | 47.00 | 48.00 | 48.00 (1.00) | 56.67 (1.53) | 61.00 |
| | ArcMemo-PS | 49.00 | 49.00 | 49.00 | 49.00 (0.00) | 59.00 (0.00) | 63.00 |
| 1 | baseline | 57.00 | 55.00 | 61.00 | 57.67 (3.06) | 66.33 (4.04) | 71.00 |
| | cheatsheet | 56.00 | 56.00 | 57.00 | 56.33 (0.58) | 65.33 (1.15) | 70.00 |
| | ArcMemo-OE | 59.00 | 55.00 | 56.00 | 56.67 (2.08) | 65.67 (1.53) | 70.00 |
| | ArcMemo-PS | 58.00 | 60.00 | 55.00 | 57.67 (2.52) | 67.00 (1.73) | 72.00 |
| 2 | baseline | 59.00 | 59.00 | 65.00 | 61.00 (3.46) | 69.00 (2.65) | 73.00 |
| | cheatsheet | 65.00 | 61.00 | 61.00 | 62.33 (2.31) | 71.33 (1.53) | 76.00 |
| | ArcMemo-OE | 62.00 | 59.00 | 61.00 | 60.67 (1.53) | 67.67 (2.52) | 71.00 |
| | ArcMemo-PS | 60.00 | 66.00 | 58.00 | 61.33 (4.16) | 70.33 (3.06) | 75.00 |

Table 5: **Strict Scoring**. While the official evaluation scheme allows different test cases solved by different attempts to be ensembled, in contrast, the strict scoring regime only marks a puzzle as solved if a single attempt (generated program) solves all test cases.

| | Individual Run Scores | | | Oracle@k Scores | | | Puzzles Improved |
|---|---|---|---|---|---|---|---|
| Iteration | run0 | run1 | run2 | k=1 | **k=2** (Official Score) | k=3 | |
| 0 | 171.50 | 185.00 | 184.50 | 180.33 (7.65) | 219.50 (1.80) | 237.00 | |
| + ArcMemo-PS | 186.00 | 185.50 | 182.00 | 184.50 (2.18) | 225.83 (2.57) | 246.50 | **36** |
| 1 | 222.00 | 221.50 | 231.00 | 224.83 (5.35) | 259.00 (5.68) | 274.50 | |
| + ArcMemo-PS | 228.00 | 225.50 | 218.00 | 223.83 (5.20) | 262.00 (5.29) | 279.50 | **22** |
| 2 | 238.50 | 242.00 | 250.00 | 243.50 (5.89) | 274.33 (5.03) | 287.00 | |
| + ArcMemo-PS | 246.00 | 251.00 | 236.00 | 244.33 (7.64) | 277.67 (7.01) | 292.50 | **21** |

Table 6: Full Evaluation Split Results (400 puzzles, base model: o4-mini). "Puzzles Improved" refers to the number of puzzles where the ArcMemo setting outscores the Baseline.

**Concept Specificity Investigation**   As part of the development of our concept representations, we conducted small-scale experiments using tiny (n=10) subsets of the validation split to investigate the level of detail needed for concepts to be useful. Manually writing maximum detail situation-suggestion style concepts and using iteratively LLM-summarized versions (for a total of 5 levels of specificity for each concept), we found the reasonably expected result that higher levels of specificity correspond to better solve rates. Maximum specificity solved 4/10 with a generally decreasing pattern with lower specificity. While this result was entirely expected, the main goal of the experiment was to determine how much we could compromise on concept detail to improve the retrieval aspect. There is a tension between retrieval and puzzle solving in the sense that puzzle solving benefits from more detailed suggestions, but retrieval struggles to apply them to new situations when the concept's relevant situation is so precisely defined.

**Embedding-based Retrieval Experiments**   Early in testing, we investigated standard embedding-based retrieval approaches. Following the ArcMemo-OE setting, we used a VLM-generated caption to query a vector database of ArcMemo-OE-style concept embeddings. While computationally cheap compared to autoregressive generation (especially so when leveraging long-form reasoning), we saw generally poor retrieval results. Using OpenAI's embeddings API and an o3-mini puzzle-solving backbone proved ineffective, lowering the score from 0.26 to 0.22, marking a 15% reduction. Qualitative analysis found that retrieved concepts were largely irrelevant to the puzzle and seemed to over-index on lexical similarity. Moreover, a small set of broadly defined (non-specific) concepts was found to be grossly overrepresented.

| | Individual Run Scores | | | Oracle@k Scores | | | Puzzles Improved |
|---|---|---|---|---|---|---|---|
| Model Name | run0 | run1 | run2 | k=1 | **k=2** (Official Score) | k=3 | |
| Kimi K2 Thinking | 36.00 | 28.00 | 19.00 | 27.67 (8.50) | 37.33 (3.79) | 42.00 | |
| + ArcMemo-PS | 38.00 | 28.00 | 22.00 | 29.33 (8.08) | 40.00 (4.58) | 46.00 | **11** |
| Sonnet-4 | 44.50 | 45.50 | 46.50 | 45.50 (1.00) | 54.17 (0.58) | 57.50 | |
| + ArcMemo-PS | 44.00 | 45.00 | 38.50 | 42.50 (3.50) | 52.50 (1.80) | 57.00 | **7** |
| DeepSeek R1 | 26.00 | 20.50 | 12.00 | 19.50 (7.05) | 26.33 (4.01) | 30.50 | |
| + ArcMemo-PS | 18.50 | 26.00 | 23.50 | 22.67 (3.82) | 30.67 (0.76) | 35.00 | **12** |

Table 7: Additional Model Results (100-puzzle subset). "Puzzles Improved" refers to the number of puzzles where the ArcMemo setting outscores the Baseline.

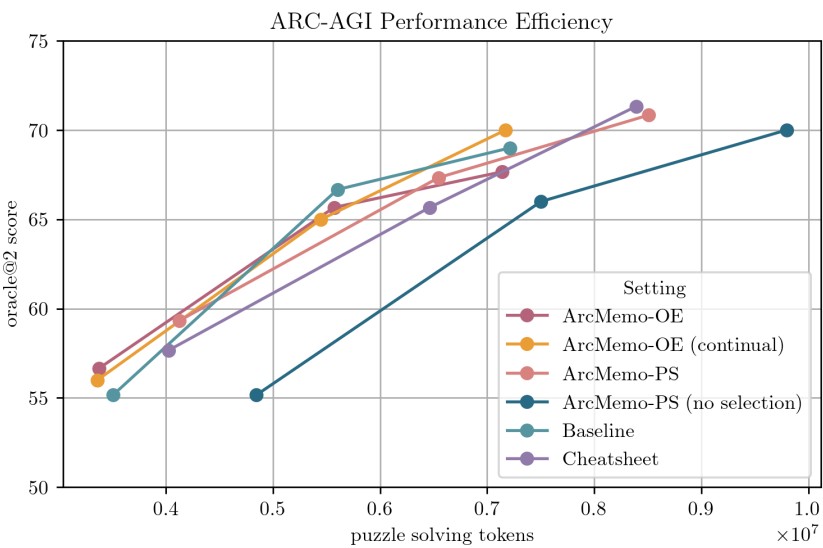

Figure 4: Token Efficiency Plot: Comparing various settings' official score (oracle@2) on the public validation subset to the tokens used by the reasoning model for strictly puzzle solving.

# D  LIMITATIONS AND FUTURE WORK

## D.1  LIMITATIONS

Following Akyürek et al. (2025), our evaluation focuses on a 100-task subset of ARC-AGI-1, selected to make large-scale experimentation feasible under compute and cost constraints. This choice was further motivated by the substantial sampling variance we observed—identical prompts can yield noticeably different scores, requiring multiple runs to obtain stable estimates. As a result, our study concentrates on a relatively small frontier of puzzles where memory augmentation can yield new solves, limiting the absolute magnitude of observable gains.

In addition, ARC-AGI-2 was released mid-project as a harder successor benchmark, with state-of-the-art performance still below 30%. Re-running all baselines and experiments on ARC-AGI-2 was infeasible within scope due to cost and time constraints, but extending memory-based approaches to ARC-AGI-2 remains an important direction for future work.

## D.2  FUTURE WORK

Several directions follow naturally from this work. One is to broaden evaluation to larger and more challenging puzzle sets, including those unsolved by the baseline, to test both new solve potential and consistency on borderline cases. Another is to explore order-robust update strategies and curriculum

designs that reduce sensitivity to problem ordering while preserving inter-problem learning benefits. Beyond these extensions, a key avenue is hierarchical abstraction: moving from additive concept updates toward consolidation operations that restructure the memory across multiple experiences. Finally, our current design choices for representation and retrieval were made under cost constraints; richer optimization may yield substantial improvements.

To facilitate further research, **we release a data resource**: hand-annotated concepts for difficult puzzles, together with a configurable puzzle synthesis pipeline. This resource is intended to support isolated evaluation of concept representations and selection mechanisms, as well as development of methods for concept abstraction. We hope it provides a useful foundation for continued exploration of abstract reasoning memory.

## E  ADDITIONAL RELATED WORK

**Knowledge Base Augmented LLMs for Factuality.**    Early efforts to enhance language models with external memory primarily targeted knowledge-intensive tasks. Retrieval-Augmented Generation (RAG) introduced the idea of retrieving documents from an external store based on embedding similarity to augment generation (Lewis et al., 2021). Follow-up work like KnowledGPT incorporated formal knowledge bases, allowing LLMs to execute structured queries through "Program-of-Thought" prompting for multi-hop reasoning (Wang et al., 2023b). More recently, MemLLM proposed inline memory operations—allowing the model to read from and write to memory directly during generation—enabling continual adaptation without re-training (Modarressi et al., 2024).

**Embedding Space Memory.**    Another stream of work aims to overcome the limited context window of transformers through architectural interventions that compress information from long sequences into continuous space. Memorizing Transformers stored latent key-value vectors from past inputs and retrieved them via a kNN mechanism to be attended to alongside the current context (Wu et al., 2022). The Recurrent Memory Transformer extended this by incorporating a writable memory updated over time, making memory content responsive to ongoing computation (Bulatov et al., 2022). He et al. (2024b) blended these ideas by applying a moving average to update memory slots, allowing for both similarity-based retrieval and continual adaptation He et al. (2024a). MemGPT introduced a hierarchical memory system with a working (RAM-like) memory and a larger archival memory, simulating long-term information management in a more structured fashion (Packer et al., 2024).

**Parameter-Free Test-time Learning.**    Test-time learning (sometimes referred to as online learning) describes methods that update predictions at inference time Bottou & LeCun (2003; 2005). However, trends towards (1) massive model sizes and (2) black box models accessed through APIs make continually updating parameters computationally impractical or downright impossible. In light of this, LLM test-time learning research sought to pursue parameter-free adaptation by using LLMs' remarkable capacity for instruction following and self-reflection. Methods such as Reflexion (Shinn et al., 2023), Self-Refine (Madaan et al., 2023), and Self-RAG (Asai et al., 2023), *inter alia* attempt to correct mistakes at test-time by way of self-reflection. Follow-up work (Huang et al., 2024; Kamoi et al., 2024) finds that self-correction methods depend on the availability of some extrinsic feedback or verification mechanism to reliably improve end performance and that these methods produce dubious results absent these grounded sources of feedback. Still, self-reflection remains a key component of self-evolving agents (Gao et al., 2025).

**Memory for Agentic Skill Acquisition.**    Memory also plays a crucial role in LLM-based agents. Park et al. (2023) structured memory as a stream of episodic observations and distilled high-level summaries from them to guide decision-making. Contextual Experience Replay discretizes trajectories into "experiences" (containing environment dynamics from a trajectory and skills storing actions from said trajectory) and retrieves relevant blocks to guide new episodes (Liu et al., 2025). HiAgent organizes working memory (within a single query) hierarchically by subgoals, summarizing and replacing traces to improve long-horizon efficiency (Hu et al., 2024). External-memory systems such as Mem0 and MemP manage a persistent store with explicit add–update–prune operations (MemP casts control as an MDP), yielding gains on dialogue and tool-use/planning tasks (Chhikara et al., 2025; Fang et al., 2025). MemP in particular is a concurrent work that also demonstrates the efficacy of abstraction, although it targets rewriting action sequences as step sequences in explicitly

agentic settings, contrasting with ArcMemo's general problem-solving focus. Voyager uses LLMs to autonomously acquire and store skills while exploring Minecraft, with a growing library of reusable code snippets written by the agent itself Wang et al. (2023a). Other systems, like the ResearchAgent, use memory initialized from domain-specific corpora, such as scientific papers, to ground query responses in high-utility prior knowledge (Baek et al., 2024).

**Abstractions from Reflection.** Other recent and concurrent works incorporate the related idea of using reflections from solution traces towards different goals and mechanisms. RLAD (Qu et al., 2025) jointly trains an abstractor model to generate new abstractions per problem for a solver model. NuRL (Chen et al., 2025) generates problem-specific hints $(h_i)$ from ground truth question-answer pairs $(q_i, a_i)$ to augment a dataset. Didolkar et al. (2025) summarizes behaviours for math reasoning, using embedding-based retrieval, and adding a fine-tuning stage on resulting traces. In contrast, ArcMemo builds an explicit store of concepts. Our mechanism is model agnostic, gradient-free, and lightweight. We especially emphasize abstraction and modularity to promote reuse, and selection-aware design to better handle unfamiliar domains appropriate for the continual learning setting.

# F PROMPTS

## F.1 PSEUDOCODE PREPROCESSING

```
# Introduction
Consider a class of "ARC" puzzles where each puzzle has a hidden
↪  transformation rule that maps input grids to output grids. Each
↪  puzzle presents several input-output grid pairs as reference examples
↪  and the task is to predict the transformation rule. Grids are 2D
↪  numpy integer arrays with integers representing colors. 0 represents
↪  black and usually serves as the background.

We are trying to learn from previous puzzle solutions to improve our
↪  puzzle solving capabilities. Your task is to analyze a puzzle
↪  solution, rewrite it as pseudocode that can more easily be abstracted
↪  into concepts, and finally write a one-liner summary of the
↪  transformation rule. Here, a concept can encode any of:
(a) grid manipulation: an operation that directly impacts the output grid
(b) helper routine: specialized logic for parameterizing more abstract
↪  operations
(c) criteria: checked properties/overall logic for conditional operations
(d) structure: shapes to look for in the pixel grids

# Instructions
Pseudocode:
- write the pseudocode translation inside <pseudocode> and </pseudocode>
↪  tags
- be concise without compromising correctness
- reuse function names/operations from the examples
- focus on broader ideas compared to implementation details
- the full solution might use `find_connected_components` to pick out
↪  objects, the output of these are numpy grids containing only the
↪  object-- implicitly storing attributes, we prefer your pseudocode to
↪  instead use a grid object type with explicit attributes like
↪  .color/.colors/.shape (returns a binary
↪  mask)/.height/.width/.size/.position/etc.

Summary:
- write a one-liner summary of the transformation rule inside <summary>
↪  and </summary> tags

# Examples
{examples}

# Concepts Examples
```

```
972   Here are concepts that were extracted from the examples. Bear in mind
973   ↪   that your generated pseudocode should enable abstracting concepts
974   ↪   such as the following:
975   {concepts}
976
977   # Your Puzzle Soution
978   Analyze, abstract into pseudocode, and summarize the following puzzle
979   ↪   solution:
980   ```python
981   {solution}
982   ```
```

## F.2   CONCEPT ABSTRACTION

```
# Introduction
Consider a class of "ARC" puzzles where each puzzle has a hidden
↪   transformation rule that maps input grids to output grids. Each
↪   puzzle presents several input-output grid pairs as reference examples
↪   and the task is to predict the transformation rule. Grids are 2D
↪   numpy integer arrays with integers representing colors. 0 represents
↪   black and usually serves as the background.

We are trying to learn from previously solved puzzles to help solve more
↪   puzzles in the future. Your task is to analyze a puzzle solution's
↪   pseudocode and abstract reusable concepts. Here, a concept can encode
↪   one of the following:
1. routine: routines from the solution program, which can either directly
↪   output a grid, or prepare handle some intermediate processing
2. structure: a class of visual entities (not program data structures)
↪   that can be seen in an individual pixel grid. The visual entity
↪   should be general, perhaps by being defined by their function or in
↪   relation to an operation.

Following a functional programming philosophy, we promote reusability
↪   through composition by parameterizing routines:
- We also encourage passing specialized logic (other routines) as
↪   parameters to higher order routines
- Each routine is typed-- it specifies output typing in addition to a
↪   list of parameter specifications
- We allow custom types to be defined with the format "name :=
↪   definition"
- Often custom types are `Callable` that specify some pluggable
↪   operations

The overarching goal is to help future puzzle solving which boils down to
↪   2 problems:
1. determining what the transformation is (by examining input/output
↪   examples)
2. implementing the transformation in code

- These concepts are meant to compactly encode ideas from solved puzzles
↪   and help puzzle solving by remembering what operations could be
↪   involved and how they might interact with each other via typing.
- Parameterization and composition via higher order routines/pluggable
↪   operations helps ensure concepts are reusable and not overly specific
↪   to one puzzle
- But to specifically address the 2 main problems, we also annotate
↪   concepts with:
    1. relevance cues: suggestion of what to look for at puzzle solving
    ↪   time that would indicate this concept is potentially relevant.
        - can either describe (1) things to look for in the puzzle grids
        ↪   or (2) related concepts that might implicate this one
```

```
1026              - we primarily care about annotating cues for grid manipulation
1027              ↪  routines and structures where there are concrete things to
1028              ↪  look for in the reference grids.
1029          2. implementation notes: suggestions on how to implement the idea
1030          ↪  programmatically.
1031
1032      # Examples
          {examples}
1033
1034      # Concept Repository
1035      Here is the current concept repository that you should check for reuse:
          {concept_list}
1036
1037      # Your Puzzle Solution
1038      Abstract the following solution into a concept list:
          ```
1039
1040      {summary}
          {pseudocode}
1041      ```
1042
1043      # Instructions
1044      - Format your final concept list inside a fenced yaml markdown block
1045      ↪  (have first line = "```yaml" and last line = "```")
          - Feel free to think before writing your final response
1046      - Each concept entry can have the following fields
1047          concept: routine or structure name
1048          kind: routine | structure
1049          routine_subtype: string specifying either "grid manipulation",
1050          ↪  "intermediate", or the name of a Callable custom type.
              output_typing: the output type
1051          parameters: a list of parameter specifications (dictionary with
1052          ↪  fields: name, typing, description)
              description: string elaborating on the concept if not completely self
1053          ↪  evident from the name
1054          implementation: a list of notes that provide suggestions on how to
1055          ↪  implement this concept programmatically
1056          cues: a list of notes suggesting what to look for at puzzle solving
1057          ↪  time that would suggest this concept is relevant (e.g. what to
1058          ↪  look for in example input/output grids, or other concepts being
1059          ↪  considered that may also implicate this concept)
1060      - Typing fields should be similar to python type annotations but may
1061      ↪  include ARC puzzle specific terminology (e.g. `list[object]`, or
1062      ↪  `Callable[[object], color]`). New custom types may be defined by
          ↪  filling out a typing field with `name := definition`
1063      - IMPORTANT: Reuse concepts and parameter names/types whenever possible
1064          - reusing a concept or parameter means reusing its exact name
1065          - check existing concepts/parameter names in the `Concept Repository`
1066          ↪  section below
              - when reusing a concept:
1067              - it need not be redefined-- a list entry of `concept: name` is
1068              ↪  sufficient
1069              - however you may also choose to extend it (introduce new list
1070              ↪  entries for the parameter list, the implementation notes, or
1071              ↪  the relevance cues which are merged with the existing
1072              ↪  entries). This involves specifying the same concept name as
1073              ↪  the existing one to be extended.
          - Avoid coupling the concepts too closely with the exact solution
1074      ↪  implementation.
1075          - often times, there is a broader idea that can be implemented with
1076          ↪  roundabout methods easier to express in code
              - we should record the broader idea in our concept memory but record
1077          ↪  the easier implementation in the implementation notes
1078      - Distinct concepts must have different names
1079      - Output valid yaml
              - avoid putting colons in entries where we're expecting a list[str]
```

```
1080        - be careful about multiline strings
1081    - IMPORTANT: No need to encode every implementation detail from the
1082    ↪    solution into a concept, focus on the important ideas, logic, and
1083    ↪    structure.
1084        - ultimately we want concepts to be reusable, so if something seems
1085        ↪    overly specific and unlikely to reappear in other puzzles, do not
            ↪    add it
1086
1087
1088
```

## F.3    CONCEPT SELECTION

```
1091
1092    # Introduction
        Consider a class of "ARC" puzzles where each puzzle has a hidden
1093    ↪    transformation rule that maps input grids to output grids. Each
1094    ↪    puzzle presents several input-output grid pairs as reference examples
1095    ↪    and solving the puzzle means predicting the transformation rule.
        ↪    Grids are 2D numpy integer arrays with integers representing colors.
1096    ↪    0 represents black and should be treated as the background.
1097
1098    Your task is to analyze a puzzle's reference examples, examine a set of
1099    ↪    concepts recorded from previously solved puzzles, and determine which
1100    ↪    concepts are relevant to this puzzle. Your selected concepts and
1101    ↪    notes will be used for the puzzle solving phase, so emphasize problem
1102    ↪    solving helpfulness.
1103
1104    # Concepts from Previously Solved Puzzles
1105    We recorded concepts about structures and routines we observed in
        ↪    previously solved puzzles. These concepts may or may not be relevant
1106    ↪    to this puzzle, but they provide useful context to show examples of
1107    ↪    what structures may appear in the grids, what operations may be used,
1108    ↪    and how they might be composed. Concepts are annotated with fields
1109    ↪    like:
        - cues: (short for "relevance cues"), what to look for that might
1110    ↪    indicate this concept is relevant in this puzzle
1111    - implementation: notes on how this concept was implemented in past
        ↪    solution programs
1112    - output typing: what the output of this routine is (e.g. a grid, a list,
1113    ↪    a number, a bool, etc.)
1114    - parameters: a list of parameters that describe ways the concept may
        ↪    vary
1115    We also have some recommendations on how to approach problem solving with
1116    ↪    these concepts in mind:
1117    - We label the grid manipulation routines separately-- these directly
        ↪    affect the grids so they are easier to spot (along with structure
1118    ↪    concepts)
1119    - You might try to first identify which grid manipulation operations are
        ↪    used, then investigate their parameters
1120    - The non-grid manipulation routines might describe ways we've seen
1121    ↪    previous puzzles set parameters, so you can look to these for
        ↪    inspiration
1122    - There may not be exact matches to this list, so we encourage you to
1123    ↪    think about variations, novel ways to recombine existing ideas, as
        ↪    well as completely new concepts
1124    - These concepts and this approach are only suggestions, use them as you
1125    ↪    see fit
1126
1127    {concepts}
1128
1129    # Instructions
1130    Identify which concepts could be relevant to the given puzzle.
1131    - We suggest first investigating more "visible" concepts first (e.g.
1132    ↪    structures and grid manipulation routines)
1133
```

```
- After identifying these concepts, you can investigate what
↪  logic/criteria/intermediate routines might be useful for these
↪  initially identified concepts
- You can also select any other concept you think might be relevant even
↪  if it's not directly related to the grid manipulation routines
- Write your final selection of concepts as a yaml formatted list of
↪  concept names
- To allow us to match your selection to the concepts we have, please use
↪  the exact concept names as they appear in the above concept list
- Write your answer inside a markdown yaml code block (i.e. be sure to
↪  have "```yaml" in the line before your code and "```" in the line
↪  after your list)
- Here is a formatting example:
```yaml
- line drawing
- intersection of lines
...
```

# Your Given Puzzle
Analyze the following puzzle:
{puzzle}
```

## G EXAMPLE CONCEPTS

### G.1 ARCMEMO-OE

```
situation: When every row in the input grid is identical or shows uniform
↪  behavior.
suggestion: Instead of looking for complex spatial rearrangements by row,
↪  focus on per-column operations. It may indicate that the
↪  transformation is applied column by column, such as a substitution or
↪  mapping rule.

situation: When each column appears to transform its digits (or colors)
↪  in a consistent but distinct way.
suggestion: Consider that each column might have its own mapping rule--a
↪  substitution cipher. Check if specific input digits in a column
↪  always map to the same output digits, and validate this rule across
↪  different examples.

situation: When the output grid is a uniformly scaled-up version of the
↪  input, especially when the output dimensions are an integer multiple
↪  of the input dimensions.
suggestion: Check if the output can be divided into equal-sized blocks
↪  that correspond one-to-one with the cells of the input. Investigate
↪  whether each cell in the input acts as a trigger--replicating a
↪  specific pattern or full copy of the input into the corresponding
↪  block--when it's nonzero, and leaving it blank otherwise.
```

### G.2 ARCMEMO-PS

```
## structure concepts
- concept: split grid
  description: where the grid is split into multiple regions that are
  ↪  treated as distinct
  cues:
    - divider lines (of any color) that span the grid and partition it
    ↪  into regions
    - visually distinct regions or objects that are processed separately,
    ↪  often separated by colored or black lines, columns, or rows
```

```
          - color-based regions where pixels of a certain color only appear
          ↪  within specific sections of the grid

  - concept: rectangle object
    description: an object whose shape is a rectangle, often used for bars,
    ↪  highlights, or as a region to fill
    cues:
      - if the output or solution logic depends on the number of distinct
      ↪  items (e.g., colors, objects) or the frequency of a color in
      ↪  regions or objects
    implementation:
      - use len() on the sequence

## grid manipulation routines
  - concept: recolor object
    description: recolor an object based on a color scheme
    output_typing: grid | object
    parameters:
      - object (type: object): the object or pixel to recolor
      - color scheme (type: dict[color, color]): mapping from original
      ↪  color to new color
      - object | object | position - the object or pixel to recolor
      - color scheme | dict[color, color] - mapping from original color to
      ↪  new color
      - color (type: color): the color to assign to the object
    cues:
      - if structures or objects from the input grid appear in the output
      ↪  grid with different colors
      - if all pixels of one color in the output grid are replaced with
      ↪  another color (systematic color swap)
      - if regions or objects are recolored based on a property (e.g.,
      ↪  size, position, or matching a specific pixel)
      - if the output grid contains regions filled with a new color not
      ↪  present in the input
      - if background or other colors are systematically replaced in the
      ↪  output grid
      - if the output grid is visually similar to the input except for a
      ↪  color swap
      - if colored pixels or regions appear in the output grid at positions
      ↪  that were black or missing in the input, especially adjacent to
      ↪  colored pixels or aligned with features of input objects
      - if a region or contiguous area is uniformly recolored, possibly
      ↪  based on a pixel in a special position (e.g., corner, interior of
      ↪  a container)
      - if an input pixel or object is removed or replaced with background
      ↪  color in the output grid (deletion by recoloring)
      - if a special color (e.g., grey or yellow) appears at a position
      ↪  determined by the interaction or meeting of input objects or
      ↪  paths
    implementation:
      - for a single pixel, set grid[position] = color
      - after drawing operations that may overwrite a pixel, restore that
      ↪  pixel to its intended color
      - for each pixel, if its color is in the color scheme, replace it
      ↪  with the mapped color
      - for a region or object, iterate over its pixels and set each to the
      ↪  target color or according to the color scheme
      - for a region, if a pixel is background (e.g., black), set it to the
      ↪  specified color; otherwise, leave it unchanged
      - at special positions (e.g., meeting points or intersections), set
      ↪  the pixel to a designated color (e.g., grey) instead of either
      ↪  input color

  - concept: repeat color sequence in rows
```

```
    description: fill each row of a grid (starting from start_row) with
    ↪  colors from a sequence, cycling through the sequence as needed
    output_typing: grid
    parameters:
      - grid (type: grid): the grid whose rows to fill
      - start_row (type: int): the index of the first row to fill
      - color_sequence (type: list[color]): the sequence of colors to
      ↪  repeat in filling rows
    cues:
      - presence of a plus-shaped pattern formed by a row and column of the
      ↪  same color in the grid
      - output grid is created by drawing a filled row and column
      ↪  intersecting at a specific position

## color selection
- concept: pick only color from region
    description: select the only non-background color present in a region;
    ↪  assumes the region contains exactly one color besides the
    ↪  background (e.g., black)
    output_typing: color
    parameters:
      - region (type: grid): the region grid to extract the color from
```