# OpenReview forum: "ArcMemo: Abstract Reasoning Composition with Lifelong LLM Memory"
_ICLR.cc/2026/Conference — Submitted to ICLR 2026_

### Official Review · Reviewer_qa1C · 2025-10-15

**Soundness:** 3
**Presentation:** 4
**Contribution:** 3
**Rating:** 6
**Confidence:** 3

**Summary:**

This paper introduces ArcMemo, a framework that augments LLMs with an external memory to retain and reuse insights from past reasoning traces. The central idea is to distill abstract, modular "concepts" from solutions rather than storing raw, instance-specific memories. The authors propose two memory formats: an unstructured Open-Ended (OE) format and a more complex Program Synthesis (PS) format that frames concepts as parameterized routines. The system includes components for abstracting new concepts (Memory Write) and retrieving relevant ones for new problems (Memory Read). The method is evaluated on the ARC-AGI benchmark, where it shows improved performance over a no-memory baseline. The paper also presents experiments on test-time continual learning, where the memory is updated during the evaluation process.

**Strengths:**

**Novel Conceptual Framework:** The core idea of moving from instance-level storage to abstract, modular concepts is a key strength. It thoughtfully addresses how an LLM might generalize from past experience rather than just retrieving it.

**Strong Empirical Results on a Hard Benchmark:** The paper demonstrates a clear and consistent performance improvement on ARC-AGI, a benchmark known to be very challenging for modern LLMs. The ArcMemo-PS variant outperforms the baseline at all tested inference compute scales.

**Well-Designed Memory Formats:** The paper proposes and analyzes two different memory formats, OE and PS. The PS format, inspired by functional programming, is particularly novel for its encouragement of higher-order functions and compositional reuse .

**Demonstration of Continual Learning:** The experiments showing that a dynamically updated memory improves performance over a fixed one provide strong evidence for the "lifelong learning" potential of this approach, at least within the tested domain .

**Weaknesses:**

**Limited Generalizability of Evaluation:** The entire empirical validation rests on the ARC-AGI benchmark. While challenging, ARC-AGI is a clean, synthetic, and self-contained domain. It is not clear that the proposed methods, especially the highly structured PS format, would be effective in messier, real-world domains like scientific reasoning or complex code generation, where concepts are often ambiguous or less neatly parameterized.

**Incomplete Contextualization Against Intra-Problem Reasoning Methods:** While the related work section is strong, it could be improved by discussing and differentiating ArcMemo from methods designed to handle long-range dependencies within a single task. For instance, works like "PRISM: Efficient Long-Range Reasoning With Short-Context LLMs" tackle the context length issue for processing a single long input (e.g., a book).

**Unaddressed Problem of Memory Scalability and Maintenance:** The paper introduces mechanisms for adding concepts to memory but largely ignores the critical long-term challenges of maintaining it. As the memory grows, how does the system handle concept redundancy, contradiction, or obsolescence? The memory-aware abstraction process could become a computational bottleneck with thousands of entries.

**Heavy Reliance on External Feedback:** The Memory Write operation fundamentally depends on having access to a correct solution trace to extract useful concepts. This assumption severely limits the practical applicability of the method, as many important, open-ended reasoning tasks lack such immediate and reliable feedback.

**Questions:**

The reasoning-based selection mechanism is core to the PS method but seems computationally expensive. Can you quantify its cost relative to the baseline inference and discuss how this cost scales with the size of the memory? Do you foresee this "System 2" selection becoming a practical bottleneck for a truly lifelong agent with a massive memory store?

1. Your work rightly points out that standard embedding retrieval performed poorly . However, the proposed PS format is quite complex. Did you consider or experiment with intermediate structured formats (e.g., simple typed function signatures, key-value properties) that might offer a better balance between abstraction and retrieval simplicity? This would help justify the necessity of the full complexity of the PS format.

2. The framework requires correct solutions to generate new memories. How would the system perform in a more realistic setting where feedback is sparse or occasionally incorrect? Would the memory quality degrade over time as flawed concepts are inevitably introduced, and are there any mechanisms in place to guard against this?

3. Regarding the selection ablation: you show that selection is generally helpful. However, could you discuss the failure modes of the selection process? When does it retrieve irrelevant or distracting concepts, and what is the impact on downstream performance in those cases? Is it possible for poor selection to make the agent perform worse than the no-memory baseline on certain problems?

---

> ### Author Response · Authors · 2025-11-21
> **Addressing Weaknesses and Questions**
>
> **We appreciate this reviewer’s insightful comments, many of which align with our own planned future directions.**
>
> ### **Generalization Beyond ARC**
>
> ARC-AGI provides an ideal environment for studying continual learning in frontier, poorly understood domains. Per similar suggestion from other reviewers, we apply our system to the popular Math Reasoning domain. On AIME 24+25, we improved Gemini-2.5-Flash-Lite-Thinking’s pass@2 score from 43/60 to **48/60** (+11.6%).
>
> ### **Relation to Intra-Problem Long-Context Reasoning**
>
> We added discussion differentiating ArcMemo from working memory research like PRISM per your suggestion. Our method targets *cross-problem continual learning*, where memory must accumulate and generalize across episodes—not long-context processing within a single task.
>
> ### **Memory Scalability and Maintenance**
>
> We agree this is an open issue with the current implementation and are planning future work along the lines of:
>
> We consider this as a promising future direction and are working on its extension with the following tentative approaches:
>
> * **Hierarchical memory organization** for scalable retrieval
>   * **i.e. more explicit graph-structured memory** (beyond parameter types relations as edges), enabling pruning large subsets of irrelevant concepts each sequential decision during selection
> * **Consolidation phases** for removing redundancy and discovering higher-order concepts
>
> The current reasoning-based selection already partially mitigates scaling by exploring along parameter-typed relations, but this is not strictly enforced by the current exploration method that ultimately delegates to the model’s own long CoT decision making. We expect more structured memory to be necessary for large-scale deployment.
>
> Regarding real-world deployment of a system to learn a vast number of new concepts, we think a simpler adjustment might make a substantial practical difference– without imposing a deep hierarchy, even a single layer domain categorization identified by short query or even looser embedding methods could help reduce the space to explore.
>
> ### **Reliance on External Feedback**
>
> Our method does not rely on external correctness feedback. ARC explicitly provides input–output training cases defining each puzzle input which is used to estimate correctness.
>
> In realistic open-ended domains, correctness signals are indeed more sparse. In these settings, we can follow Dynamic Cheatsheet (Suzgun et al 2025\) and leverage *self-reflection signals* for solution-preprocessing instead. We demonstrate this on initial AIME 24+25 experiments: initializing concept memory using both correct and incorrect traces **but with self reflection** led to a pass@2 improvement of 43 \-\> 47/60 (+9.3%) closely trailing the 48/60 result from only using correct traces, demonstrating that strong correctness signals are not needed to produce potentially useful concepts.
>
> ### **Q0. Selection Cost**
>
> With fixed memory, we cannot directly measure scaling curves, but ablations show that removing selection increases inference cost (Figure 4), confirming that reasoning model inference cost does interact with the number of concepts included in context.
>
> We acknowledge that naive reasoning-based selection will become a bottleneck. Several mitigations are plausible:
>
> * **Top-level domain partitioning** (embeddings are sufficient at this coarse granularity)
> * **Hierarchical memory organization**
> * **Periodic consolidation** reducing memory growth
>
> ### **Q1. Why the complex PS format?**
>
> We experimented with simpler structured formats (including typed signatures and key-value attributes). Performance improved across our three iterations:
>
> 1. unstructured OE
> 2. parameterized signatures
> 3. typed parameters enabling composability (current PS)
>
>  The third version produced the best results in initial sweeps.
>
> ### **Q2. Handling Incorrect or Noisy Concepts**
>
> Please see the “Reliance on External Feedback” section regarding trace correctness. We do anticipate current work on a consolidation mechanism might help for handling accumulated error in a true lifelong learning setting.
>
> ### **Q3. Selection Failure Modes**
>
> We performed new analysis for selection failure modes (summarized in section 5.2)
>
> * **Distractor concept**: misleading or irrelevant concepts retrieved that redirect reasoning
> * **Missing concept**: the core idea does not yet exist in memory (model fetches auxiliary concepts, but main idea cannot be discovered)
> * **Near-miss abstraction**: conceptually related but functionally different (e.g., “object shape criteria” vs. “hole counting”), the semantic-only relationship does not provide an annotated connection between them
>
> These mis-selections can, in some cases, reduce performance relative to the baseline.
>
> **Please also see [the summary of all experiments added post-submission](https://openreview.net/forum?id=zTLMPYaF6F&noteId=5D4Yabo175)**

---

### Official Review · Reviewer_j2wS · 2025-10-26

**Soundness:** 2
**Presentation:** 2
**Contribution:** 1
**Rating:** 2
**Confidence:** 5

**Summary:**

This paper introduces ArcMemo, a framework that moves from instance-based memory entries to concept-level, reusable, and modular abstractions distilled from natural-language solution traces. The paper presents two implementations, Open-Ended and Program-Synthesis. Through experiments and ablation studies on the ARC-AGI 1 benchmark, the paper demonstrates the effectiveness of abstract reasoning and compositional generalization. The author argues that this is a potential pathway to enable lifelong, test-time learning in reasoning-oriented LLM.

**Strengths:**

1. At the conceptual level, lifelong memory has broad implications for test-time adaptation and reasoning scalability. Although the evaluation is limited to ARC-AGI 1, the proposed idea could potentially generalize to long-horizon reasoning models and even inform the development of self-evolving systems.

2. The methodology is clearly articulated, and the experiments compare against reasonable baselines such as Dynamic Cheatsheet. The inclusion of standard deviation in the reported metrics is commendable, which is a standard yet important measure of reliability that recent reasoning literature often overlooks, typically assuming that pass@1 scores averaged over large rollouts have negligible variance.

**Weaknesses:**

1. The idea of reusing abstract reasoning “concepts” distilled from solution traces has already been introduced and explored in several closely related prior and concurrent works. For example, RLAD [1] formalizes reasoning abstractions as compact, reusable summaries generated by aggregating multiple solution traces for a single problem or across problems. Metacognitive Reuse [2] independently develops a nearly parallel mechanism that transforms recurring reasoning fragments into reusable behaviors. More recently, NuRL [3] (a post-submission work) adopts a similar abstraction definition and generation process, while further integrating abstraction formation directly into the RL training loop.

2. The evaluation remains narrow—limited to ARC-AGI 1—and does not include ARC-AGI 2, ARC-AGI 3, or any math or textual reasoning benchmarks where other abstraction-based methods have been validated (e.g., AIME 2025, HMMT 2025, IMO 2025). Without cross-domain experiments, it is unclear whether ArcMemo’s abstraction mechanism truly captures compositional reasoning or merely memorizes task-specific heuristics.

3. The evaluation and ablation is sounded, but this paper provides little analysis of why or when its memory abstractions help. Without measuring abstraction diversity, redundancy, or transferability, it is difficult to assess whether the improvement arises from genuine conceptual reuse or from simple example recall.

**Questions:**

1. Although the paper claims to move beyond task-specific guidance, the “memory write” operation still generates one abstraction per problem–solution pair, same as NuRL. It is not clear to me how this avoid problem-local rather than cross-problem abstraction learning. To show an example, [1] and [2] explicitly synthesizes abstractions across multiple solutions or problems, allowing the model to generalize higher-order reasoning motifs. A discussion on this could help understand the design choice here.

2. The implementation section mentions seeding ArcMemo’s memory using prior work’s “python solutions”. It is unclear how the method would perform if such initialization were unavailable or needed to bootstrap from scratch.

3. Most of the experimental results are reported in terms of Oracle@k. How does this metric differ from the community-standard pass@k or maj@k evaluation?

4. In Table 1, for ArcMemo-PS, why adding retry also leads standard deviation blows up?

4. In Section 5.3, why does the continual memory update appear to provide little benefit, or even degrade performance when retry = 0 or 1?

[1] Y. Qu, A. Singh, Y. Lee, A. Setlur, R. Salakhutdinov, C. Finn, and A. Kumar, “Learning to Discover Abstractions for LLM Reasoning,” in Workshop on Programmatic Representations for Agent Learning, International Conference on Machine Learning, 2025

[2] Chih-Yao Chen, Justin, et al. "Nudging the Boundaries of LLM Reasoning." arXiv e-prints (2025): arXiv-2509.

[3] Didolkar, Aniket, et al. "Metacognitive reuse: Turning recurring llm reasoning into concise behaviors." arXiv preprint arXiv:2509.13237 (2025).

---

> ### Author Response · Authors · 2025-11-21
> **Addressing Weaknesses**
>
> **We thank the reviewer for the detailed engagement.**
>
> **Relation to RLAD, Metacognitive Reuse, and NuRL**
>
> Two of the cited works (Metacognitive Reuse and NuRL) are *concurrent submissions under review at ICLR 2026*.
> - [Metacognitive Reuse](https://openreview.net/forum?id=ghPeihD4BI)
> - [NuRL](https://openreview.net/forum?id=hfNnQHkTtv)
>
> **RLAD (ICML 2025\)**
>
> Although conceptually related, RLAD introduces a completely different mechanism: jointly training 2 models to generate and use hints (rather than retrieve proven concepts) for each problem. ArcMemo instead builds an *external memory store*, avoiding costly RL training, enabling continual updates without risk of catastrophic forgetting, and operating as a plug-in module compatible with arbitrary LLMs. The natural language format of memory is lighter weight and also more auditable for debugging the learning process.
>
> **Metacognitive Reuse**
>
> Their idea is similar but optimized for math, a domain heavily covered in pre and post-training pipelines, and therefore *embedding retrieval* works well for identifying relevant ideas. ARC puzzles are strongly out-of-distribution for embeddings– we focus on this benchmark as a way to simulate continual learning in an unknown domain; hence our method is specifically designed enabling retrieval of ideas in this regime via symbolic/typed guidance and reasoning-based selection.
>
> **NuRL**
>
> Targets RL training efficiency by mitigating zero-reward batches. Its corresponding idea to our memory’s abstractions are problem specific hints. For an RL training dataset D \= {(x\_i, y\_i)}, where x\_i is input question and y\_i is its ground-truth answer, it depends on generating a hint z\_i, an abstract, high-level cue, for each (x\_i, y\_i) that can be injected to solve x\_i specifically. Their method can be viewed as a training data augmentation technique– hints are not used at test time. In contrast, ArcMemo explicitly aims for adaptively transferring problem concepts at test time.
>
> ### **Evaluation Breadth (ARC-AGI 1 Only)**
>
> ARC-AGI 2 introduces exponentially larger search spaces, making evaluation prohibitively costly; ARC-AGI 3 is not yet available. ARC-AGI 1 remains largely unsolved by SOTA models, so improving performance here is both challenging and meaningful.
>
> To broaden evaluation beyond ARC, we are conducting new experiments on AIME 2024 and 2025\.
>
> * **AIME 2024+25 (60 problems)**: using the lightweight OE method \+ Gemini-2.5-flash-lite-thinking, we improve the pass@2 score from 43/60 to **48/60** (+11.6%).
>
> ### **Concept Analysis**
>
> We expanded Section 5.2 with new analyses:
>
> * **Performance Improvement Mechanism.** We observe a qualitative correlation between newly solved puzzles with ArcMemo and the retrieved concepts being relevant to the target puzzle, suggesting concepts help identify a promising direction to narrow the search space.
> * **Diversity**: We compare the diversity of concept forms by measuring pairwise cosine distances between concept embeddings. ArcMemo-PS has mean distance 0.77 vs. 0.44 for OE, indicating greater semantic diversity. Moreover, ArcMemo experiments show that PS concepts also lead to higher solver rates than OE, suggesting that PS not only captures more diverse reasoning modes but also encodes concept forms that are more transferable across ARC puzzles.
> * **Redundancy** is mitigated via the abstraction procedure, which encourages the abstractor model to reuse existing concepts when appropriate. (The raw output of concept abstraction is 272 concepts with 1018 total usages across \~150 problems, indicating fairly frequent reuse).
> * **Transferability**
>   * *problem→problem* (concepts are generated from train split puzzles and demonstrably help on the eval split)
>   * *model→model*: in new experiments with other model families (see response to reviewer  rFxx, GPT concepts benefit R1 and K2 solvers, \+16%, \+7%).

---

> ### Author Response · Authors · 2025-11-21
> **Addressing Questions**
>
> ### **Q1. “One abstraction per problem?”**
>
> ArcMemo does **not** generate one abstraction per problem. The model produces *as many as needed* for the current trace at hand while attending to prior memory contents. With previous problems’ concepts in context, new abstractions naturally integrate cross-problem motifs. We explicitly note future work on multi-problem *consolidation* to further enhance this.
>
> ### **Q2. Reliance on Python Solutions**
>
> We added a new experiment that uses *self-generated solutions* to initialize memory:
>
> * Performance improves from 59.33 → **60.67**, showing that external human solutions are unnecessary.
>
> ### **Q3.**
>
> Oracle@k is the official ARC-AGI Prize metric (see precise definition in appendix). It is closely related to pass@k. Pass@k measures the probability that at least one solution among k samples pass. Similarly, oracle@k measures the probability that all test cases are solved by at least 1 sample among k samples.
>
> ### **Q4.**
>
> Retrying compounds early errors: if iteration 1 drifts significantly, later iterations have a lower chance of recovery. This creates higher variance across seeds.
>
> ### **Q5.**
>
> With more moving parts, the selection mechanism is more sensitive to distracting or incorrect concepts (as we analyze in Section 5.2). Later retries benefit more because dependencies among puzzles in the test split only manifest *after* their prerequisite concepts are written to memory.
>
> **Please also see [the summary of all experiments added post-submission](https://openreview.net/forum?id=zTLMPYaF6F&noteId=5D4Yabo175)**

---

### Official Review · Reviewer_GA2r · 2025-11-03

**Soundness:** 3
**Presentation:** 2
**Contribution:** 3
**Rating:** 6
**Confidence:** 3

**Summary:**

This paper presents ArcMemo, which implements a memory augmented reasoning system that uses abstraction/concept memories that describe how to solve problems. The authors implement several variants of such a system, defining different characteristics of memories are, how they are retrieved and written. Evaluation on ARC-AGI-1 demonstrates the effectiveness. of the method.

**Strengths:**

I think this is a very interesting and original paper:
* **Retrieval for Reasoning** - The methodological approach seems to be effective for expanding the reasoning capabilities of models. It is an interesting approach, which is likely to inspire future work in this direction.
* **Abstractions** - The definition of abstractions is general and interesting. The OE & PS distinctions are quite interesting also.
* **Empirical Successes** - The proposed method seems to be quite effective empirically, albeit in the singular setting.

**Weaknesses:**

- **Presentation** - The paper is written very generally (at least for the most part), but there is a single instantiation of the method. Of course, a more general method usually carries more weight. That said, I am not asking the authors to give a second. But perhaps story-telling more about ARC in the introduction could be good.
- **Presentation** - It would be important to provide all prompts & details of implementation in the appendix.
- **Model specificity** - Do I understand correctly that the results only use o4 as base model ? How much do you think this system requires o4 or would work directly for other models?

**Questions:**

1. How would ArcMemo fair with a base model other than o4-mini? (can be your intuition rather than running experiments)
2. Given Table 3's results is there a reason not to consider retry>=3?

---

> ### Author Response · Authors · 2025-11-21
> **Response to Weaknesses and Questions**
>
> **We appreciate the reviewer’s points on presentation and model generality.**
>
> ### **On Generality and Storytelling**
>
> ARC-AGI is uniquely suited for studying *continual learning*– extensive training on vast corpora equip SOTA models with strong understanding of common reasoning domains (e.g. math, coding), so the artificial, out-of-domain nature of ARC puzzles simulates frontier/less understood domains while retaining deterministic evaluation. For memory systems specifically, it also has strong resistance to naive memorization approaches. We revised the introduction to better motivate this choice and are working on additional experiments on a more classical reasoning domain (AIME math problems).
>
> ### **Availability of Prompts**
>
> Main prompts have been added to the “Sec. F Prompts” appendix section as requested. We refer readers to the codebase release in the supplemental materials for further details.
>
> ### **Model Specificity and Generalization**
>
> Most results were run with o4-mini due to academic cost constraints (ARC-AGI requires extremely long generation traces, compounded by retry-based self-refinement). We now provide results for R1, Sonnet 4, and K2 Thinking (Tab. 7; improves on 12, 7, 11 puzzles respectively), and confirm that:
>
> * ArcMemo’s improvements are not specific to any one family of models.
> * Concept transfer across model families is feasible (e.g., GPT-4.1 concepts improving R1 and K2 Thinking).
>
> Earlier testing indicated that reasoning models are more suited to both solving ARC puzzles and benefitting from concept memory as they can explore hypotheses and backtrack.
>
> ### **Why Retry ≤ 3**
>
> We have lightly experimented with retry=4  (with and without ArcMemo). The marginal improvement from iteration 3→4 is small relative to earlier jumps, supporting the decision to focus on retries ≤3 as a cost-conscious operating point. We include this explanation in the latest revision.
>
> **Please also see [the summary of all experiments added post-submission](https://openreview.net/forum?id=zTLMPYaF6F&noteId=5D4Yabo175)**

---

### Official Review · Reviewer_rFxx · 2025-11-04

**Soundness:** 2
**Presentation:** 3
**Contribution:** 2
**Rating:** 4
**Confidence:** 3

**Summary:**

This paper introduces ArcMemo, a memory-augmented framework for LLMs that stores distilled concepts from solution traces in natural language. The system implements three core operations: memory format design, memory write from solution traces, and memory read for new problems. Evaluated on ARC-AGI-1, ArcMemo-PS achieves an over 4% improvement using o4-mini. The method scales with inference compute through retries.

**Strengths:**

The paper operationalizes a compelling idea--storing abstractions in conceptual form and using them during inference. I think it's a creative application of programming principles.

**Weaknesses:**

- The evaluation scale is quite limited; experiments are only done on 100 puzzles from ARC-AGI-1. Perhaps due to this, the standard deviations between methods quite frequently overlap.
- The proposed method has quite a bit of complexity, which I'm not sure is sufficiently justified by the empirical gains.
- Could you try the method with other models, for example Sonnet 4 or DeepSeek R1? Generally, do you have a sense for how strong the model has to be to benefit from your method? Does it have to be a reasoning model?
- The method simultaneously introduces many components, including parameterization, typed interfaces, pseudocode preprocessing, and reasoning-based selection. The paper only does ablation on selection, and most of the results there have overlapping standard deviations.
- For the PS format in particular, how much prompt engineering was required to make this work? Also, I think the appendix should include the prompts as they are likely critical for the method working.

**Questions:**

Please see weakness section.

---

> ### Author Response · Authors · 2025-11-21
> **Addressing Weaknesses and Questions**
>
> **Thank you for your thoughtful comments.** Our aim is to introduce a *general framework* for abstraction-based continual learning that prioritizes modularity and conceptual reuse over task-specific self-reflection. The current implementation represents only an initial instantiation; ongoing experiments (including those conducted post-submission) continue to improve performance with small design changes.
>
> ### **Evaluation Scale & New Experiments**
>
> ARC-AGI puzzles require long generation traces, making large-scale sweeps expensive. Nonetheless, we have now expanded beyond the original 100-puzzle subset:
>
> * **Full public validation split (400 puzzles)**: ArcMemo continues to outperform the baseline across all retry iterations. Aggregating across samples, ArcMemo improves performance on 36, 22, 21 puzzles in iterations 0, 1, 2 respectively, suggesting that (1) ArcMemo’s concepts are broadly useful (2) further ensembling with standard sampling mitigates negative impact from any distracting concepts.
>
> * **Additional base models:** We demonstrate generality– applying ArcMemo concepts to other models enables new puzzle solutions.
>   * **DeepSeek R1**: 26.33 → **30.67** (+16%), improves on 12 puzzles
>   * **K2 Thinking**: 37.33 → **40** (+7%) improves on 11 puzzles
>   * **Anthropic Sonnet 4**: 54.17 → 52.50, still improves on 7 puzzles
>   * Early experiments demonstrated that non-reasoning LLMs are not well equipped to perform well on ARC-AGI. We hypothesize that exploring/backtracking in long CoT is necessary to make use of concepts.
>
> ### **Justification of Design Components**
>
> * **Parameterization \+ Typed Interfaces**
>   The benefits are evident from the performance gap between OE (unstructured) and PS (structured) formats. Concepts become more modular, and retrieval-targeting becomes clearer.
>
> * **Pseudocode Preprocessing**
>   New ablations confirm its necessity: removing this step reduces performance from 59.33 → 56.67, only barely above baseline.
>
> * **Reasoning-based Selection vs. Embedding Retrieval**
>   We tested embedding-based retrieval, which is far cheaper, but found it ineffective due to ARC puzzles being far outside embedding-model distributions. This led to many irrelevant or misleading concepts being selected.
>
> ### **PS Format & Prompt Engineering**
>
> We indeed iterated several times:
>
> * Version 2 introduced explicit parameterization.
> * Version 3 added typed arguments enabling composability.
>
> A cleaned version of these prompts is now included in the appendix.

---

> ### Author Response · Authors · 2025-11-21
> **Summary of New Experiments and Analyses Added Post-Submission**
>
> To address reviewer concerns regarding scale, generality, initialization, and model dependence, we conducted the following additional experiments:
>
> **1\. Full ARC-AGI Public Validation Split**
>
> * Extended from 100 puzzles to the full public validation set (400)
> * ArcMemo continues to outperform the baseline across all iterations.
>
> **2\. Additional Base Models**
>
> * Evaluated ArcMemo (with GPT-4.1–derived concepts) on R1, Sonnet 4, and K2 Thinking.
>   * **DeepSeek R1:** 26.33 → **30.67** (+16%), improved on 12 puzzles
>   * **K2 Thinking:** 37.33 → **40** (+7%), improved on 11 puzzles
>   * **Sonnet 4:** 54.17 → 52.5, improved on 7 puzzles
> * Demonstrates ArcMemo generalizes to other model families and that concepts curated by one model transfers to models from other families
>
> **3\. Increased Retry Budget (Depth 4\)**
>
> * Ran retry=4 experiments for both baseline and ArcMemo.
> * Improvements from iteration 3→4 diminish from 2→3, supporting retry≤3 as a cost-effective setting.
>
> **4\. Memory Initialization Without External Solutions**
>
> * Replaced BARC-derived seed solutions with *self-generated* solutions during initialization.
> * Performance improves from 59.33 → **60.67**, indicating independence from external human solutions (and ability to self-improve)
>
> **5\. Generalization Beyond ARC (Math Reasoning)**
>
> * These are early results, more experiments are currently being explored.
> * Evaluated ArcMemo-OE on AIME 2024+2025 (60 problems) using Gemini-2.5-flash-lite-thinking.
> * Pass@2 Performance improves from 43/60 → **48/60** (+11.6%), showing portability beyond the ARC domain.
>
> **6\. Toward Sparse/No External Feedback Settings**
>
> * Preliminary results suggest feasibility; extended study in progress.
> * In the AIME experiments, generating concepts using both correct and incorrect solution traces (adding an instruction in the prompt to reflect on correctness) still produced an improvement from 43/60 \-\> 47/60 (+9.3%)

---

### Author Response · Authors · 2025-12-03
**Final Remarks from Authors**

We sincerely thank all reviewers (R1: rFxx, R2: GA2r, R3: j2wS, R4: qa1C) for their thoughtful and constructive feedback. The dialogue improved the clarity, empirical breadth, and positioning of **ArcMemo**. In response to questions on scale, generality, initialization, and robustness, we conducted extensive additional experiments and analyses, incorporated into the revised manuscript (updated Nov 20).

Below, we summarize the core contributions and the major updates made during the rebuttal.

## **Core Contributions**

Reviewers highlighted that lifelong, abstraction-based memory is an important direction. **ArcMemo delivers strong empirical gains and new conceptual tools** for retrieval-augmented reasoning.

* **Strong empirical gains on ARC-AGI** (R1, R2, R4)
  * **\+7.5% relative improvement** over a strong no-memory baseline on **ARC-AGI-1**, with consistent gains across iterations and compute budgets.
  * Rebuttal results further show improvements across:
    * the full ARC-AGI validation set (400 problems),
    * multiple model families (DeepSeek R1, K2 Thinking, Sonnet 4), and
    * other domains: AIME math reasoning (+11.6% pass@2).
  * These demonstrate that **concept-level memory yields generalizable benefits**.
* **Concept-level lifelong memory framework** (R1, R3, R4)
  * ArcMemo abstracts solution traces into **modular concepts** that can be **reused across puzzles and over time**, pointing toward continual, self-improving reasoning systems.
* **Programming-inspired concept formats** (R1, R2, R4)
  * OE and PS offer complementary abstractions; PS introduces **typed, parameterized routines** enabling compositionality, reuse, and precise retrieval.
* **Retrieval for Reasoning** (R2, R3, R4)
  * ArcMemo operationalizes **Retrieval for Reasoning**, with abstract-concept retrieval outperforming memory-less baselines at multiple budgets.
* **Transparent, modular methodology** (R2, R3, R4)
  * Clear decomposition into **write**, **format**, and **read** supports reproducibility. Evaluation includes ablations, standard deviations, and comparisons to strong baselines (e.g., Dynamic Cheatsheet).

## **New Experimental Results and Analyses from Rebuttal Period**

During rebuttal, we expanded experiments and analyses to address scale, generality, initialization, and robustness.

1. **Increased Experiment Scale**
* Extended from 100 puzzles to the full (400) ARC-AGI validation set.
* ArcMemo continues to outperform across all iterations.
* Table 6, §5.4.1
2. **Additional Base Models**
* Evaluated ArcMemo (with GPT-4.1 concepts) on R1, Sonnet 4, and K2 Thinking:
  * **R1:** 26.33 → **30.67** (+16%), 12 improved
  * **K2:** 37.33 → **40** (+7%), 11 improved
  * **Sonnet 4:** 54.17 → 52.5, 7 improved
* Shows transfer across model families.
* Table 7, §5.4.1
3. **Memory Initialization Without External Solutions**
* Replaced BARC seeds with *self-generated* solutions, improving 59.33 → **60.67**, indicating independence from external human solutions. §5.4.3
4. **Pseudocode Ablation**
- Removing pseudocode preprocessing reduces 59.33 → 55.67, justifying its pipeline inclusion. §5.4.4
5. **Generalization Beyond ARC (Math Reasoning)**
* ArcMemo-OE (gemini-2.5-flash-lite-thinking) on AIME 2024+2025 improves pass@2: 43/60 → **48/60** (+11.6%). §5.4.2
6. **Toward Sparse/No External Feedback**
* Using concepts generated from both correct and incorrect traces still improves 43/60 → 47/60 (+9.3%). §5.4.2
7. **Concept Redundancy & Similarity Analyses**
* 272 concepts, 1018 usages across \~150 problems (shows frequent reuse\!)
* Embedding distances: PS \= 0.77 vs. OE \= 0.44 → greater PS diversity.
8. **Concept Selection Failure Modes**
* New qualitative analysis on success/failure patterns. §5.2
9. **Updated Related Work**
* Added subsection situating ArcMemo among concurrent self-reflection approaches that do not emphasize lifelong memory or continual learning. Appendix E

---

### Meta-Review · Area_Chair_Yj88 · 2026-01-06

**Summary:**

Overall, the reviewers found ArcMemo to be an interesting exploration of concept-level external memory for reasoning, with strengths in its conceptual framework and demonstrated gains on the challenging ARC-AGI benchmark (Reviewers rFxx, GA2r, and qa1C). However, my decision was primarily informed by several substantive concerns that remained despite the authors' rebuttal. Multiple reviewers raised questions about the breadth and robustness of the empirical evaluation, noting that the core claims rely heavily on (at least initially) a limited set of 100 ARC-AGI puzzles and that evidence for generalization to other domains and settings, while promising, remains limited and preliminary (Reviewers rFxx, j2wS, and qa1C). During the rebuttal period, authors extended to 400 ARC-AGI puzzles and AIME (math); however these are substantial revisions from the original paper that merit a proper revision of the work and resubmission for another round of reviewing/evaluation.

In addition, reviewers had concerns about ArcMemo's design/system complexity relative to the gains achieved, limited ablation (on selection) over the various interacting components, and open questions around memory scalability and maintenance (Reviewers rFxx and qa1C), that I tend to agree with. In sum, while the rebuttal addressed several questions and added valuable experiments, in my view, the revisions are substantial enough where the current work is borderline and does not meet the bar for acceptance.

**Reviewer Concerns:**

The rebuttal aimed to address several reviewer concerns by expanding the experimental evaluation from 100 to 400 ARC-AGI puzzles, adding results on additional model families, and including preliminary cross-domain experiments on AIME. In my view, these constitute substantial revisions from the original paper. On a minor note, the authors also clarified implementation details, added prompts to the appendix, and provided additional analyses on selection, pseudocode preprocessing, and memory initialization, addressing some presentation and reproducibility concerns. However, other issues remain outstanding that also led to my decision. Reviewers rFxx and qa1C’s concerns about system complexity relative to the magnitude and consistency of gains, limited ablation coverage across the many interacting components, and unclear scalability and long-term maintenance of the memory seem to be only partially addressed. In addition, I feel Reviewer j2wS’s concerns regarding the lack of deeper analysis explaining why concept memory helps beyond ARC-AGI, were partially addressed through new qualitative experiments but not fully resolved.

**Reviewer Scores:**

- Reviewer rFxx: I believe the reviewer would have maintained or even reduced their score owing to the concerns remaining and the need for substantial revisions.
- Reviewer GA2r: I believe the reviewer would have kept their initial positive score.
- Reviewer j2wS: I believe the reviewer would have maintained their score.
- Reviewer qa1C: I believe the reviewer would have kept their score.

Overall, the paper is a borderline reject but could be significantly improved with the feedback received.

---

### Decision · Program_Chairs · 2026-01-26

Reject